



# Influence of photochemical aging on light absorption of atmospheric black carbon and aerosol single scattering albedo

Xuezhe Xu[1,2], Weixiong Zhao[1], Xiaodong Qian[1,2], Shuo Wang[1,2], Bo Fang[1], Qilei Zhang[1,2], Weijun Zhang[1,2,3], Dean S. Venables[4], Weidong Chen[5], Yong Huang[6,7], Xueliang Deng[6,7], BiwenWu[6,7], Xinfeng
Lin[7,8], Sen Zhao[7,8], Yingxiang Tong[7,8]

[1]Laboratory of Atmospheric Physico-Chemistry, Anhui Institute of Optics and Fine Mechanics, Chinese Academy of Sciences, Hefei, 230031, Anhui, China
[2]Graduate School, University of Science and Technology of China, Hefei, 230026, Anhui, China
[3]School of Environmental Science and Optoelectronic Technology, University of Science and
Technology of China, Hefei, 230026, Anhui, China
[4]School of Chemistry and Environmental Research Institute, University College Cork, Cork, Ireland
[5]Laboratoire de Physicochimie de l'Atmosphère, Université du Littoral Côte d'Opale, 59140 Dunkerque, France
[6]Anhui Institute of Meteorological Science, Hefei, 230031, Anhui, China
[7]Shouxian National Climatology Observatory, Shouxian, 232200, Anhui, China
[8]Anhui Shouxian Meteorological Bureau, Shouxian, 232200, Anhui, China

*Correspondence to*: Weixiong Zhao (wxzhao@aiofm.ac.cn) and Weijun Zhang (wjzhang@aiofm.ac.cn)

**Abstract.** Coating enhancement of black carbon (BC) light absorption ($E_{abs}$) is a large uncertainty in
modelling direct radiative forcing (DRF) by BC. Reported $E_{abs}$ values after atmospheric aging vary widely and the mechanisms responsible for enhancing BC absorption remain elusive. Here, we report on the direct field measurement of size-resolved mixing state, $E_{abs}$ and aerosol single scattering albedo (SSA) at $\lambda = 532$ nm at a rural site in East China from June to July 2016. Strong diurnal variability of $E_{abs}$, SSA, and $O_x$ ($O_x = NO_2 + O_3$, a proxy for atmospheric photochemical aging) was observed. A
three-stage absorption enhancement process for collapsed semispherical to fully compact spherical morphology BC with photochemical aging was suggested. For $O_x$ below 35 ppbv, $E_{abs}$ increased slowly with $O_x$ mixing ratio and ranged from 2.0 to 2.2 (with a growth rate of ~ 0.03 ppbv$^{-1}$). $E_{abs}$ was stable ($E_{abs} = 2.26 \pm 0.06$) between 35 to 50 ppbv $O_x$. Thirdly, for $O_x$ levels above 50 ppbv, $E_{abs}$ grew rapidly from 2.3 to 2.8 (at a growth rate of ~ 0.18 ppbv$^{-1}$). A method that combined $E_{abs}$ and SSA was
developed to retrieve the fraction contribution of BC absorption ($f_{BC}$), lensing driven enhancement ($f_{Lens}$), as well as the fractional contribution of coating absorption (fraction absorption contribution



($f_{Shell}$), the coated shell diameter ($D_{Shell}$) and the imaginary part of the complex refractive index (CRI) of the shell ($k_{Shell}$)). Parameterization of $E_{abs}$ and SSA captures much of the influence of BC coating and the particle absorption, and provides a plausible new method to better constrain the contribution of BC to the DRF. In our measurements at this site, the absorption amplification depended mainly on the

coating thickness and the absorption of coating materials. The lensing driven enhancement was reduced by light absorption of the shell. Our observations highlight the crucial role of photochemical processes in modifying the absorption of BC-containing particles. One implication of these findings is that the contribution of light-absorbing organic compounds (Brown carbon, BrC) at longer aging time should be included in climate models.

**1 Introduction**

Black carbon (BC) is the most efficient light absorbing component of atmospheric aerosols (Jacobson, 2001; Moffet and Prather, 2009; Cappa et al., 2012) and plays an important role in the global climate system (Ramanathan and Carmichael, 2008; Bond et al., 2013). However, accurately constraining the direct radiative forcing (DRF) of BC is a challenge owing to the discrepancy between observed and

modeled estimates of BC light absorption (Gustafsson and Ramanathan, 2016). For example, a recent study has shown that the improved model estimated DRF of BC ($+0.21$ $Wm^{-1}$) by including BC absorption enhancement and separately treat the aging and physical properties of fossil-fuel and biomass-burning BC was about 3 times lower than the values reported in the Intergovernmental Panel on Climate Change (IPCC) 5th assessment report ($+0.6$ $Wm^{-2}$), which suggested an overestimation of

BC lifetime and an incorrect absorption attribution of light-absorbing organic compounds (Brown carbon, BrC) (Wang et al., 2014).

BC particles are produced from incomplete combustion of fossil fuels, biofuels and residual biomass (Novakov et al., 2003; Bond et al., 2004; Bond et al., 2007). Freshly emitted BC is mainly externally mixed and occurs in fractal-like agglomerates. Atmospheric BC particles undergo several

aging processes, including coagulation with other particles, condensation of vapors onto surfaces, and chemical oxidation (Slowik et al., 2004; Zhang et al., 2008; Petzold et al., 2013). Individual BC particles become coated (i.e., internally mixed) with sulfate, ammonium, organics, nitrate and water





(Bond and Bergstrom, 2006; Cheng et al., 2006; Schwarz et al., 2008; Ervens et al., 2010; Zaveri et al., 2010). Aging processes dramatically change the morphology, hygroscopicity, and mixing state of BC-containing particles, thereby altering their optical properties and the magnitude of their contribution to climate forcing (Jacobson, 2001; Bond et al., 2006; Schwarz et al., 2008; Zhang et al., 2008).

The light absorption enhancement of BC particles caused by coating is quantified by $E_{abs}$, the ratio of the absorption coefficients of coated and bare BC. $E_{abs}$ introduces a large uncertainty in the DRF of BC, which is the second most important contributor to global warming (Jacobson 2001; Liu et al., 2015). Current models simply adopt a constant enhancement value ($\sim$ 1.5 or 2) for the calculation of DRF of BC (Cappa et al., 2012; Bond et al., 2013; Wang et al., 2014). In contrast, reported $E_{abs}$ values vary

widely (Peng et al., 2016; Liu et al., 2017). Field measurements along the California coast and ground site in Sacramento (California) (Cappa et al., 2012), Shenzhen (South China) (Lan et al., 2013), the Nagoya urban area (Japan) (Nakayama et al., 2014), and urban Los Angeles (USA) (Krasowsky et al. 2016) found negligible absorption enhancement ($E_{abs} < 1.1$) and weak dependence on the extent of photochemical aging (estimated from the value of $-\log([NO_x]/[NO_y])$, where $NO_x = NO + NO_2$ and $NO_y$

includes the sum of $NO_x$ and its oxidation products (Deolal et al., 2012)). Biomass burning measurements showed an absorption enhancement of 1.7 at $\lambda = 532$ nm (Lack et al., 2012). Recent observations in Chinese cities (Peng et al., 2016; X. Cui et al., 2016; Xu et al., 2016; Chen et al., 2017; Cheng et al., 2017; Q. Wang et al., 2017) provide evidence for a higher $E_{abs}$ in polluted conditions, with values ranging from 2 to 3. The mechanisms responsible for enhancing BC absorption remain elusive

due to the complexity of the aging process and its varied sources. More studies in receptor locations with longer BC aging time are required to better constrain $E_{abs}$ (Gustafsson and Ramanathan, 2016; Boucher et al., 2016).

     In this work, the influence of photochemical aging on BC mixing state, $E_{abs}$ and aerosol single scattering albedo (SSA, $\omega$, defined as the ratio of scattering to extinction coefficient) at a rural site in

East China during the summer was studied by using a volatility Tandem Differential Mobility Analyzer (VTDMA) and a thermal denuder (TD) approach combined with a cavity enhanced albedometer operating at $\lambda = 532$ nm. In summer, $O_x$ ($O_x = O_3 + NO_2$) exhibits good correlation with secondary pollutants (Zhou et al., 214; Cevik et al., 2016; Ji et al., 2016). The concentration of $O_x$ was used as a



proxy for atmospheric photochemical aging (Hallquist et al., 2016; Q. Wang et al., 2017) in this work. We find that photochemical aging results in the growth of particle coating and higher fractions of internally mixed BC particles. A three-stage absorption enhancement process for collapsed semispherical to fully compact spherical morphology BC with photochemical aging was proposed. The

modeling and parameterization of $E_{abs}$ and SSA capture the variability of BC coating amount and the particle absorption, and provide a plausible new method to better constrain the contribution of BC to the DRF.

## 2 Experimental

### 2.1 The field site

Measurements were performed at Shouxian National Climatological Observatory (32°25'47.8"N, 116°47'38.4"E) in Anhui Province from 16 June to 23 July 2016. Shouxian County is located in China's north-south climate transition zone and is affected by the East-Asian monsoon. The new observatory is situated about 15 km south of the previous, historically important observation site (Fan et al., 2010; Li et al., 2011; Deng et al., 2012); it is a rural background site surrounded by basic farmland protection

areas and has no significant industrial pollution sources or tall buildings nearby.

Instruments were installed in a temperature-controlled room with two sample inlets about 1 m above the roof (Fig. S1 in the supplement). Each inlet consisted of one $PM_{2.5}$ cyclone (BGISCC2.654) with a 50% cut point at 2.5 μm, and was firstly dried below 40% relative humidity (RH) using a diffusion drier. The sampling rates at both inlets were controlled with mass flow controllers (MFC) and

set at 10 L min$^{-1}$. One of the inlets was used for the volatility measurements; the other inlet stream was used for the optical measurements. Trace gas pollutants such as CO, $NO_x$, $SO_2$, and $O_3$ were respectively measured by Thermo 48i, 42i, 43i, and 49i analyzer.

### 2.2 Volatility measurement

Size-resolved mixing state of BC was measured with a home-build VTDMA. The VTDMA was

structurally similar to other systems described in the literature (Cheng et al., 2009; Wehner et al., 2009; Cheng et al., 2012; Cheung et al., 2016) and comprised: (1) an electrostatic classifier (DMA, TSI 3080)



for the initial selection of mono-disperse particles; (2) a custom-built stainless steel heating tube (inner diameter of 0.77 cm, 80 cm long, and heated to $300 \pm 5$ ℃) for removing nonrefractory particulate matter; and (3) a scanning mobility particle sizer (SMPS, TSI 3936) comprising a DMA (TSI 3080) and a condensation particle counter (CPC, TSI 3776) for measuring the size distribution of the heated

sample in the range of 15 to 661 nm. Diffusion losses and the effect of multicharged particles were corrected by the instrument software. The residence time of the sample in the heating tube was about 1.2 s and is comparable with other VTDMA systems (0.3 ~ 1 s) (Brooks et al., 2002; Philippin et al., 2004; Villani et al., 2007).

### 2.3 Optical measurement

The optical properties of dry $PM_{2.5}$ particles were measured with a cavity-enhanced albedometer operating at $\lambda = 532$ nm (Zhao et al., 2014; Xu et al., 2016). The albedometer combined broad-band cavity enhanced spectroscopy (BBCES) with an integrating sphere (IS) for direct, in situ, and simultaneous measurement of extinction ($b_{ext}$) and scattering ($b_{scat}$) coefficients, thus allowing calculation of the absorption ($b_{abs}$) coefficient and SSA. Compared with our previously reported 470 nm

system (Zhao et al., 2014; Xu et al., 2016), the new 532 nm albedometer was modified by inserting a quartz tube within the IS to prevent the degradation of the IS reflectivity and to reduce the sample's residence time (Dial et al., 2010; Onasch et al., 2015). The sample volume of the albedometer was about 0.3 L and the flow rate was 1.5 L $min^{-1}$ at atmospheric pressure. With a 30 s integration time (an average of 300 individual spectra, each of 100 ms exposure time), the detection limits under ambient

aerosol loading condition for the scattering and extinction measurements were better than 0.15 and 0.12 $Mm^{-1}$, respectively. The accuracy of the instrument was evaluated with laboratory-generated, NIST traceable monodispersed polystyrene latex (PSL) spheres. During field observations, the optical system was calibrated with $N_2$, $CO_2$, and PSL every two weeks.

    The total uncertainties in extinction, scattering, absorption coefficients, and SSA measurements

were estimated to be less than 4%, 3%, 5%, and 4%, respectively. Uncertainty in extinction mainly arose from the uncertainties in mirror reflectivity ($1 - R$, ~ 1%), the ratio of cavity length to the cell length containing the air sample when the cavity mirrors were purged ($R_L$, ~ 3%), and particle losses in





the cavity (~ 2%). Uncertainty in the scattering measurement was mainly caused by uncertainties in the experimentally determined scattering calibration coefficient ($K'$, ~ 2%), particle losses in the cavity (~ 2%), and the truncated fraction of total scattering (Since most particles in the observation were smaller than 1 μm, the uncertainty associated with truncation angle was < 1%, as discussed in Sect. S2 in the

supplement). Since measurements of the extinction and scattering coefficients were of the identical sample, particle losses do not affect the SSA measurement (Zhao et al., 2014; Xu et al., 2016).

  The sampled ambient air was divided into two channels: the first channel was directly pumped into the albedometer to measure the ambient absorption coefficient ($b_{abs,ambient}$); another channel was installed with a TD (Dekati Ltd., Finland) operating at 300 ºC to evaporate semi-volatile particulate

components for measuring the absorption coefficient of rBC ($b_{abs,TD}$) (Olson et al., 2015). These two channels were switched automatically every 5 min with an electric ball valve. The flow rate of the TD was 10 L min$^{-1}$. Particle losses inside the TD are detailed discussed in Sect. S3 in the supplement, which are generally caused by diffusional and thermophoretic processes (Wehner et al., 2002; Fierz et al., 2007). The optical loss of the TD of ambient aerosol was estimated to be ~32 %. The measured $b_{abs,TD}$

was corrected with the particle losses for further calculation of the absorption enhancement ($E_{abs}$ = $b_{abs,ambient}/b_{abs,TD}$). The total uncertainty in $E_{abs}$ measurement was dominated by the uncertainty in sample losses in the TD (~ 9%).

## 3 Results and discussion

The concentrations of $PM_{1.0}$, $PM_{2.5}$ and trace pollutants (CO, $NO_2$ and $O_3$) measured at the station over

the measurement period are shown in Fig. 1. For assessing the effect of photochemical oxidation on the aerosol optical properties, the time series of the $O_x$ concentration is also shown in the figure. The corresponding meteorological conditions are shown in the supplement Fig. S8. The average ambient temperature (T), relative humidity (RH), and wind direction (WD) were 26.0 ± 3.3 °C, 90 ± 11 %, 2.0 ± 1.1 m s$^{-1}$, respectively. The prevailing winds were southerly. Generally, the low wind speed favored

accumulation of pollutants, and the RH was also quite high. The average concentrations of $PM_{2.5}$ and $PM_{1.0}$ were 28 ± 14 and 25 ± 13 μg m$^{-3}$, respectively. The 48 h backward trajectories ending at 500 m above ground level at the Shouxian site are shown in Fig. S9. The trajectories were aggregated into 5





groups after taking into account the wind direction, speed, and the geometric distance between individual trajectories (S. Wang et al., 2017). The air masses in clusters 1, 3, 4 and 5 originated from long range transport with high speeds for over 40 h. Air masses in cluster 2 originated from the vicinity of Anhui province and moved slowly. The long aging time and residence time of the air masses led to

well-aged particles before arriving at the observation site. All air masses were at relatively low altitudes (< 1500 m) and remained within the boundary layer over this two day period.

### 3.1 Size-resolved mixing state of BC

Following the approach of Philippin et al. (2004), Cheng et al. (2009), and Wehner et al. (2009), most compounds were assumed to be volatilized at 300ºC and the residual nonvolatile particles were regarded

as refractory BC (rBC). An example of the measured size distribution is shown in Fig. 2. The heated size distribution was divided into three size ranges – "high-volatility" (HV), "medium-volatility" (MV), and "low-volatility" (LV) – to calculate the number fraction of internally mixed BC particles:

$$F_{in} = N_{MV}/(N_{MV} + N_{LV}),\tag{1}$$

where $N_{MV}$ is the number concentration of MV particles and is considered as internally mixed BC. $N_{LV}$

is the number concentration of LV and is considered as externally mixed BC (Cheng et al., 2009; Wehner et al., 2009; Cheng et al., 2012; Cheung et al., 2016).

To assess the influence of atmospheric photochemical aging on the mixing state of BC, scatter plots of $F_{in}$ at different diameters and $O_x$ concentrations are shown in Fig. 3. Data points are color-coded with respect to the concentrations of CO, which is related to primary BC emission. In this work, low $F_{in}$

values tended to appear at high CO concentrations, consistent with freshly emitted BC. A positive correlation between $F_{in}$ and $O_x$ was observed for particle diameters of 150, 200, and 250 nm, respectively. The corresponding oxidation rates of $F_{in}$ were 0.08% ppb$^{-1}$, 0.12% ppb$^{-1}$, and 0.19% ppb$^{-1}$, respectively. Particles of these sizes have greater internal mixing and may be more susceptible to photochemical oxidation processes. Very recently, Q. Wang et al. (2017) reported a similar correlation

between the number fraction of thickly-coated rBC ($F_{rBC}$, the mixing state of individual rBC was measured with single particle soot photometer, SP2) and $O_x$ concentration in highly polluted megacities. The reported oxidation rates of $F_{rBC}$ were 0.58% ppb$^{-1}$ for Beijing and 0.84% ppb$^{-1}$ for Xi'an,



respectively. Photochemical aging resulted in higher amounts of internally mixed BC and a larger fraction of thickly-coated BC under more oxidizing conditions.

## 3.2 Temporal and diurnal variations of optical properties

Time series of the measured optical properties are shown in Fig. 4 and include the extinction ($b_{ext}$),

scattering ($b_{scat}$), and absorption ($b_{abs}$) coefficients, the SSA for ambient particles ($b_{ambient}$) and for particles passed through TD ($b_{TD}$), and the corresponding $E_{abs}$. The mean (and standard deviation) of $b_{ext, ambient}$, $b_{scat, ambient}$, $b_{ext, TD}$, $b_{scat, TD}$ were 92 ± 64, 81 ± 55, 12 ± 7, and 6.5 ± 4.1 Mm$^{-1}$, respectively. The scattering fraction remaining ($b_{scat,TD}/b_{scat, ambient}$) was about 0.09 ± 0.05 and indicated that most of the coating species evaporated in the TD at 300 ºC. Our value is comparable to the value (0.08 ± 0.02)

reported by Nakayama et al. (2014). The change in the morphology during heating was negligible.

The observed diurnal variation of optical parameters ($b_{ext, ambient}$, $b_{scat, ambient}$, $\omega_{ambient}$, $b_{abs, ambient}$, $b_{ext, TD}$, $b_{scat, TD}$, $b_{abs, TD}$, $\omega_{TD}$, $E_{abs}$), mass concentrations of PM$_{2.5}$, as well as the mixing ratio of CO and the photochemical oxidant ($O_x$) are shown in Fig. 5. Broadly similar diurnal patterns were observed for the extensive optical properties ($b_{ext, ambient}$, $b_{scat, ambient}$, $b_{abs, ambient}$, $b_{ext, TD}$, $b_{scat, TD}$, $b_{abs, TD}$) of ambient

particles and particles passed through the thermodenuder, and the mass concentrations of PM$_{2.5}$. A strong diurnal variation and similar diurnal patterns in $\omega_{ambient}$, $\omega_{TD}$, $E_{abs}$, and $O_x$ was observed. Patterns of the extensive optical properties and PM$_{2.5}$ indicate some local particle emissions from early morning anthropogenic activities. While these changes are radiatively significant, changes in PM$_{2.5}$ during the early daylight period are weak, suggesting that emitted particles are small and contribute little to the

overall particle mass concentration. The SSA shows that particles tend to be more strongly absorbing in early morning than later in the day; however, measured SSA values are not especially low (mean $\omega_{ambient} \geq 0.85$), consistent with the background nature of the Shouxian site. Thus, freshly emitted particles are therefore relatively unimportant at this site. CO concentrations show minor diurnal variation, consistent with the regional nature of air masses at this site.

Daytime increases in the boundary layer into the mid-afternoon are especially evident in the PM$_{2.5}$ concentration profile. In contrast, ambient scattering and extinction profiles are broadly flat over the same period, indicating more intense photochemical processing and extensive secondary aerosol





generation. The same effect is responsible for the mid-afternoon maximum in the intensive optical property $\omega_{ambient}$.

### 3.3 Influence of photochemical aging on $E_{abs}$ and SSA

SSA is one of the most relevant intensive optical properties (Jo et al., 2017) because it describes the

relative strength of the aerosol scattering and absorption capacity and is a key input parameter in climate models. Changes in particle size, morphology, chemical composition and mixing state caused by atmospheric chemical aging processes will alter SSA. Positive correlations between $\omega$, $\omega_{TD}$ and $O_x$ concentrations were observed in our measurements (Fig. 6 (a) and (b)). The increase in SSA under higher $O_x$ concentrations suggests a higher contribution from secondary aerosols formed during daytime

photochemical processing. Our result is consistent with Beijing summer observations, where SSA was linearly correlated with the mass fractions of secondary aerosols (Han et al., 2017). The increase in $\omega_{TD}$ resulted from incomplete vaporization of non-volatile constituents in the heating tube (Cheung et al., 2016) and the generation of low-volatility oxygenated organic aerosol during photochemical aging (Paciga et al., 2016).

$E_{abs}$ also rose with higher $O_x$ mixing ratios (Fig. 6 (c)), but with a different pattern compared to $\omega$ and $\omega_{TD}$. For $O_x$ larger than 50 ppbv, $E_{abs}$ grew rapidly with increasing $O_x$ (from 2.3 to 2.8, with a growth rate of ~ 0.18 ppbv$^{-1}$). Below 50 ppbv $O_x$, two regions could possibly be discerned. For $O_x$ mixing ratios below 35 ppbv, $E_{abs}$ ranged from 2.0 to 2.2 and increased slowly with the $O_x$ mixing ratio (~ 0.03 ppbv$^{-1}$). In the second region, $E_{abs}$ was unchanged (2.26 ± 0.06) for $O_x$ mixing ratios between 35

and 50 ppbv. These two regions are most likely corresponded to Peng et al.'s (2016) two-stage morphology variation mechanism, in which collapsed semispherical BC is transformed to a spherical morphology (Gustafsson and Ramanathan, 2016). Here we propose a three-stage $E_{abs}$ growth process with photochemical aging that proceeds from a collapsed semispherical BC to a fully compact spherical morphology BC.

A list of recently reported $E_{abs}$ values is shown in Table 1. The averaged and standard deviation of $E_{abs}$ value at $\lambda$ = 532 nm for this work was 2.3 ± 0.9, which agreed well with values from Boulder using the same TD method combined with photoacoustic spectrometer (PAS) (Lack et al., 2012), from





Yuncheng (X. Cui et al., 2016) and Jinan (Chen et al., 2017) using an aerosol filtration-dissolution (AFD) method, and from Beijing (Peng et al., 2016; Xu et al., 2016; Cheng et al., 2017) based on the mass absorption efficiency (MAE) method. Our result is also comparable to that reported in laboratory studies of thickly coated BC particles where $E_{abs}$ ranged from 1.8 to 2.4 (Bond et al., 2013).

Chamber study by Peng et al. (2016) suggested that the primary BC was in chain-like structure with low particle-density, then collapsed to semispherical particle. During this stage, there is no significant absorption enhancement ($E_{abs}$ ranged from 1.0 to 1.4). With continued coating growth with several hours aging in the chamber, semispherical particle was further collapsed, and finally transformed to fully compact spherical internally mixed BC particles ($E_{abs}$ increased to ~ 2.3 – 2.4)

(Gustafsson and Ramanathan, 2016). The three-stage process report here is consistent with, but an extension of, Peng et al.'s (2016) two-stage morphology variation mechanism. Photochemical aging processes lead to internal mixing and a larger coating fraction that enhances the light absorbing capacity of BC particles (Lack and Cappa, 2010). The new finding of the third stage, rapid growth of $E_{abs}$ with $O_x$ concentration, could indicate that secondary organic aerosol (SOA) includes light-absorbing organic

compounds (BrC) (Xu et al., 2016), and that BrC's overall contribution to particle absorption grows under more oxidizing conditions. As discussed in next section, we find an increase in the imaginary part of CRI of the coated shell.

### 3.4 Coating absorption and light absorption enhancement

Mie theory, which was treated as the basis of the IPCC 5th assessment report due to its computational

efficiency and applicability to radiative transfer models (Jo et al., 2017), is a powerful tool for optical data interpretation (Lack et al., 2012) and the reliability of the core-shell model has been verified in many optical closure studies (Lack et al., 2012; Ma et al., 2012; S. Liu et al., 2015; Wu et al., 2018). According to Peng et al.'s (2016) chamber study results, BC particles change to a fully compact spherical morphology in less than one day. Volatility measurements and analysis of the air masses

indicated that the atmospheric aerosol observed in summer at the rural site were well aged. A method based on single-particle core-shell Mie theory (Bohren and Huffman, 1983) was developed to interpret the proposed three-stage aging mechanism observed in this work. The modelling was based on




exploring the relationship between $E_{abs}$ and SSA to retrieve the fraction contribution of BC absorption ($f_{BC}$), lensing driven enhancement ($f_{Lens}$), coating absorption ($f_{Shell}$), as well as the coated shell diameter ($D_{Shell}$) and the imaginary part of the complex refractive index (CRI) of the shell ($k_{Shell}$).

A scatter plot of measured diurnally-averaged $E_{abs}$ and SSA for different photochemical oxidant concentrations is shown in Fig. 7. The color-coded plot shows the connection between $E_{abs}$, SSA, and atmospheric photochemistry. A linear relationship between $E_{abs}$ and SSA was observed. Both $E_{abs}$ and SSA increased under more oxidizing conditions. This can be explained by the photochemical production of coating species: with more intense photochemical aging, the fraction of internally mixed BC particles and coating thickness increased. Thickly coated BC was also observed by Q. Wang et al. (2017) under higher $O_x$ mixing ratios.

The corresponding Mie theory calculation results are shown as open circles in Fig. 7 (with further details in Sect. S6 in the supplement). Comparisons of modeling and observation $E_{abs}$ and SSA are shown as a scatter plot in supplement Fig. S11. By fixing the BC core diameter, we can retrieve information on the coating shell ($D_{Shell}$, $k_{Shell}$) and each contribution to light absorption ($f_{BC}$, $f_{Lens}$, $f_{Shell}$) under different oxidant conditions (Lack and Cappa, 2010), as shown in Fig. 8. The retrieved $D_{shell}$ ranged from 386 - 440 nm. The corresponding $D_{shell}/D_{core}$ ratio ranged from 2.41 - 2.75, within the range of values (2 - 4) reported by Wu et al. (2018). The plot of measured and modelled $E_{abs}$ with different $D_{shell}/D_{core}$ is shown in the supplement Fig. S12. The values of $k_{shell}$ ranged from 0.004 to 0.008 with a diurnal average value of 0.006 ($\pm$ 0.001). A comparison of the retrieved $k_{shell}$ with previously reported $k$ values of fresh and aged organic materials is shown in the supplement Fig. S13, which include BC, BrC aerosol production from biomass burning (BB), atmospheric humic-like substances (HULIS), Suwannee River Fulvic Acid aerosol (SRFA), and secondary organic material (SOM) produced by photo-oxidation of anthropogenic and biogenic organic precursors. The value of $k_{shell}$ reported here is comparable with those of BB aerosols (Chakrabarty et al, 2010) and SRFA (Bluvshtein et al., 2017), and is larger than those of SOM (Liu et al., 2013; P. Liu et al., 2015), HULIS (P. Liu et al., 2015) and urban BrC (Cappa et al., 2012).

A ternary plot of the fractional contribution of $f_{BC}$, $f_{Lens}$, $f_{Shell}$ is shown in the supplement Fig. S14. At the first stage of Fig. 6(c), $D_{shell}$ increased with $O_x$ concentrations, but $k_{shell}$ shown an obscure




variation with the increasing $O_x$ mixing ratios. The rise in $E_{abs}$ was mainly caused by the thicker coating. In the second stage, with constant $E_{abs}$, all the parameters ($D_{shell}$, $k_{shell}$, $f_{BC}$, $f_{Lens}$, and $f_{Shell}$) remained fairly constant. In the third stage, the coating materials became thicker and more absorbing with increasing $O_x$ concentrations. The fractional contributions of coating absorption increased from 20% to

30%, but the contributions of BC absorption and the lensing effect decreased. Our results suggest that the contribution of the lensing effect to absorption enhancement is limited (Bond et al., 2006). The lensing effect is reduced due to the greater absorption of the shell (Lack and Cappa, 2010). The change in optical properties at higher oxidant conditions imply a non-negligible contribution of absorbing secondary aerosol material to photochemistry, and should receive more attention in climate modelling

(Jo et al., 2016).

## 4 Implications for global models

The global-mean DRF by BC can be expressed as (Bond et al., 2013):

$$DRF_{BC} = E \times L \times MAC_{BC} \times AFE, \qquad (2)$$

where $E$, $L$ are the global mean emission rate and mean life time of BC, respectively. The product of $E$

and $L$ is the mean column burden of BC. $MAC_{BC}$ is the global mean absorption cross section and is equal to the ratio of the global mean absorption aerosol optical depth (AAOD) to the mass burden of BC. AFE is the global mean absorption forcing efficiency. The measured AAOD of BC can be acquired from the measured aerosol optical depth (AOD) and SSA (Bond et al., 2013; X. Wang et al., 2014):

$$AAOD_{BC} = AOD \times (1 - \omega). \qquad (3)$$

$AAOD_{BC}$ plays a critical diagnostic role in constraining model calculation (Bond et al., 2013). However, current models still underestimate AAOD (Gustafsson and Ramanathan, 2016) and the model estimated DRF by BC needs to be adjusted by a scaling factor to match atmospheric observations. The reported scaling factor derived in the comparison with AERONET observations was 1.4 to 2.8 for the GC-RT model (Wang et al., 2014), and 3.0 for the CAM5 AeroCom model (Bond et al., 2013).

After considering absorption enhancement and particle absorption (1-ω), the scaling factor acquired by comparing the measured and modeled BC AAOD can be expressed as:





$$scaling\ factor = \frac{AOD \times (1-\omega)_{measured}}{E \times L \times (E_{abs} \times MAC_{bare\ BC})}. \tag{4}$$

From the linear relationship of $E_{abs}$ and SSA obtained from section 3.3, $E_{abs}$ is about 1.9 at $\omega$ = 0.85, but increases under more scattering conditions (higher SSA), so that it is ~ 3 at $\omega$ = 0.9. If we consider both variations of $E_{abs}$ and SSA at the same time, the value of $(1 - \omega)/E_{abs}$ at $\omega$ = 0.85 will be ~ 2.4 times larger than that at $\omega$ = 0.9. We suppose that the scaling factor used to improve agreement between model and observation in current models should be considered separately in different pollutant regions or in different absorption conditions. This can also explain why the modeled AAOD values reported in Wang et al. (2014) matched well with the measured AAOD mean values in most regions, but were underestimated by a factor of two in South Africa, where biomass-burning is dominant and more absorbing. Ground in situ optical and chemical composition measurements have been widely used for testing the methodology for the retrieval of column properties. We conclude that more work needs to be done in the future under different pollutant and meteorological conditions and that parameterization of $E_{abs}$ and SSA can capture the variability of BC coating amount and the particle absorption. This may provide a plausible, new method to better reconcile model-based and observation-constrained DRF by BC.

## 5 Conclusion

In this work, the size-resolved mixing state of atmospheric BC particles, light absorption enhancement and SSA at $\lambda$ = 532 nm were measured at a rural site in East China in the summer of 2016. The volatility measurement shows that atmospheric BC particles were well-aged. A three-stage $E_{abs}$ growth process with $O_x$ concentration was proposed for collapsed semispherical to fully compact spherical BC. A single-particle core-shell Mie theory that connected $E_{abs}$ and SSA was developed to interpret the observation. Although further improvements of the calculation with size distributed BC core and coated shell may give a more complete model, the current used model with fixed BC core diameter was found to be useful in illustrating the aging process. In our summer time observations, the absorption amplification was mainly determined by the coating thickness and the absorption of coating materials.



The increase in $f_{Shell}$ highlights the crucial role of photochemical processes in modifying BC absorption, and indicates that light-absorbing organic compounds require more attention in climate modelling.

A linear relationship was observed between $E_{abs}$ and SSA. We consider that the parameterization of $E_{abs}$ and SSA is applicable to non-near-source "real" atmosphere conditions, where freshly emitted BC

particles undergo several hours or days aging processes and become well-aged. The parameterization of these two properties under different oxidant concentrations may also improve calculation of the DRF at the top of the atmosphere (TOA) (Kassianov et al., 2013; Kuang et al., 2015), especially in photochemically active seasons when clear diurnal patterns are observed (Kajii et al., 1998).

**Acknowledgements**

This research was supported by the National Natural Science Foundation of China (41330424), the Natural Science Foundation of Anhui Province (1508085J03), the Youth Innovation Promotion Association CAS (2016383), and the China Special Fund for Meteorological Research in the Public Interest (GYHY201406039).

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



**Table 1**: A survey of some field measured E$_{abs}$ values.

| Method | Location | $E_{abs}$ | Reference | Description |
|---|---|---|---|---|
| AFD | Yuncheng, China (Rural) | 2.25 ± 0.55 (678 nm) | X. Cui et al., 2016 | June-July 2014; ECOC analyzer. ($E_{abs}$ ranged from 1.4 for fresh combustion emissions to 3 for aged ambient aerosols.) |
| | Jinan, China (Urban) | 2.07 ± 0.72 (678 nm) | B. Chen et al., 2017 | February 2014; ECOC analyzer. $E_{abs}$ ~ 1.3-1.5 for fresh urban aerosols, and ~ 2-2.5 for aged aerosols. |
| MAE | Shenzhen, China (Urban) | 1.07 (532 nm) | Lan et al., 2013 | August-September 2011; Absorption coefficients at 405, 532, 781 nm were measured with PAS; rBC mass concentration was measured with SP2; MAE$_{532nm}$ = 6.5 ± 0.5 m$^2$ g$^{-1}$ (with lowest value of 6.08 m$^2$ g$^{-1}$ and highest value of 8.5 m$^2$ g$^{-1}$, respectively treated as totally externally mixed and internally mixed.); SP2 measured BC core diameter ~ 180 nm. |
| | Xi'an, China (Urban) | 1.8 (870 nm) | Q. Wang et al., 2014 | December 2012-January 2013; Light absorption was measured with PAS; rBC concentration was measured with SP2. |
| | Nanjing, China (Suburban) | 1.6 (532 nm) | F. Cui et al., 2016 | November 2012; Absorption coefficients at 405, 532, and 781 nm were measured with PAS; EC mass concentration was determined by ECOC analyzer. |
| | Beijing, China (Suburban) | 2.6-4.0 (470 nm) | X. Xu et al., 2016 | November 2014 – January 2015, for PM$_{1.0}$ particles; Absorption coefficient at 470 nm by using a cavity enhanced albedometer; EC mass concentration was determined by ECOC analyzer. |
| | Beijing, China (Urban) Houston, USA (Urban) | 2.4 (405, 532 nm) | Peng et al., 2016 | May-June 2009 in Houston, August-October 2013 in Beijing; Chamber study; Absorption coefficients at 405, 532, and 870 nm were measured with PAS. |
| | Manchester, UK (Urban) | 1.0-1.3 (532 nm) | D. Liu et al., 2017 | October-November 2014; Chamber study and open wood fire measurement; Absorption coefficients at 405, 532, and 781 nm were measured with PAS. |
| | Kanpur, India (Urban) | 1.8 (781 nm) | Thamban et al., 2017 | January-February 2015; Absorption was measured with PAS; rBC concentration was measured with SP2. |
| | Beijing, China (Urban) | 3.2-5.3 (365 nm) | Y. Cheng et al., 2017 | Comparison of water-soluble and methanol-soluble organic carbon; Theoretical investigation of $E_{abs}$. |
| | Beijing and Xi'an, China (Urban) | 1.9 (532 nm) | Q. Y. Wang et al., 2017 | February 2013, Xi'an, and February 2014, Beijing; Absorption was measured with PAS; rBC concentration was measured with SP2. |
| | Guangzhou, China (Suburban) | 1.5 ± 0.5 (550 nm) | Wu et al., 2018 | February 2012-January 2013; Light absorption was measured with an Aethalometer; EC mass concentration was determined by ECOC analyzer. |
| TD | Toronto, Canada (suburban) | 1.6-1.9 (550 nm) | Knox et al., 2009 | December 2006 to January 2007; TD operating at 340 ℃; Optical properties were measured with PAS and Aethalometer. |
| | California, USA (Rural) | 1.06 (532 nm) | Cappa et al., 2012 | June 2010; TD operating at 250 ℃; Absorption coefficients at 405 and 532 nm were measured by PAS; SP2 measured rBC core diameter ~ 174 nm. |
| | Boulder, USA (Forest fire) | 2.5 (404 nm) 1.4 (532 nm) | Lack et al., 2012 | September, 2010; TD operating at 200 ℃; Absorption coefficients at 404, 532, and 658 nm were measured with PAS; SP2 measured rBC core diameter: 140±10 nm. |
| | Nagoya, Japan (Urban) | 781 nm, TD 300 ℃ 1.10±0.09 (August) 1.02±0.11 (January) | Nakayama et al., 2014 | August 2011, January 2012; TD operating at 100, 300 and 400 ℃; Absorption coefficients at 405 and 781 nm were measured with PAS. |
| | London, UK (Rural) | 1.3 (405 nm) 1.4 (781 nm) | S. Liu et al., 2015 | February 2012; TD operating at 250 ℃; Absorption coefficients at 405 and 781 nm were measured with PAS; rBC core diameters ranged from 100 to 200 nm. |
| | Noto Peninsula, Japan (Rural) | 1.22 (781 nm, ranged from 1.07-1.38) | Ueda et al., 2016 | April-May 2013; TD operating at 300 or 400 ℃; Absorption coefficients at 405, 532, and 781 nm were measured with PAS. |
| | California, USA (Urban) | 1.03 ± 0.05 (870 nm) | Krasowsky et al., 2016 | February-March 2015; TD operating at 230 ℃; Absorption at 870 nm was measured with PAS. |
| | **Shouxian, China (Rural)** | **2.3 ± 0.9 (532 nm, ranged from 2.0-2.8)** | **This work** | June-July 2016; TD operating at 300 ℃; Absorption at 532 nm was measured with a cavity enhanced albedometer. |



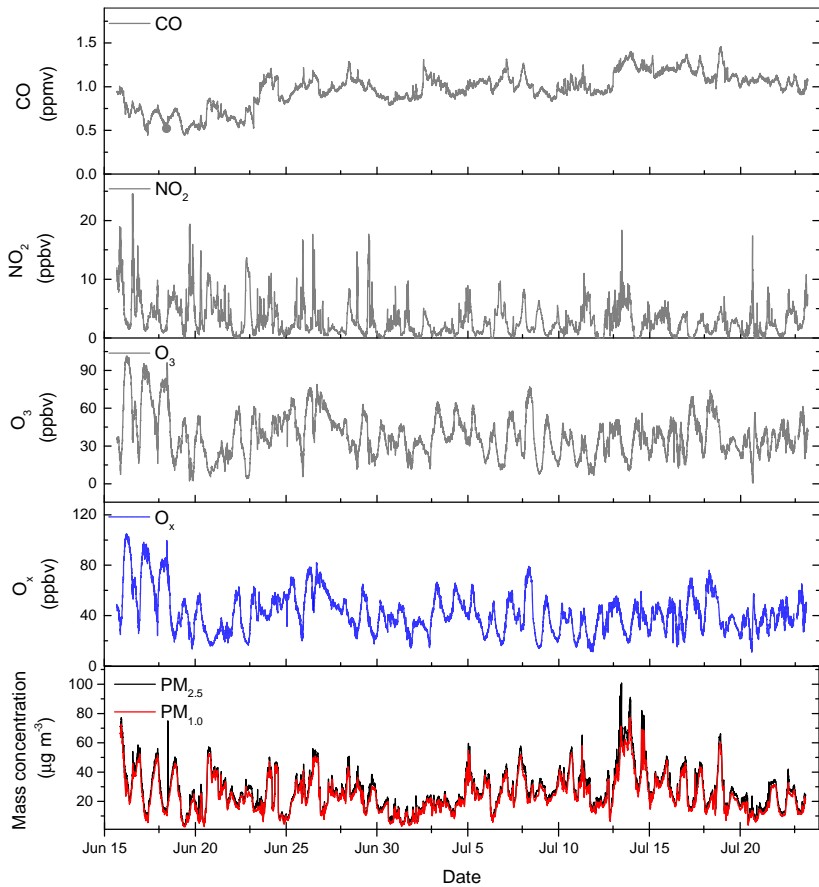

**Figure 1:** Time series of CO, NO$_2$, O$_3$, and O$_x$ (O$_3$ + NO$_2$) concentrations, as well as the concentrations of PM$_{2.5}$ and PM$_{1.0}$ during the measurement period.




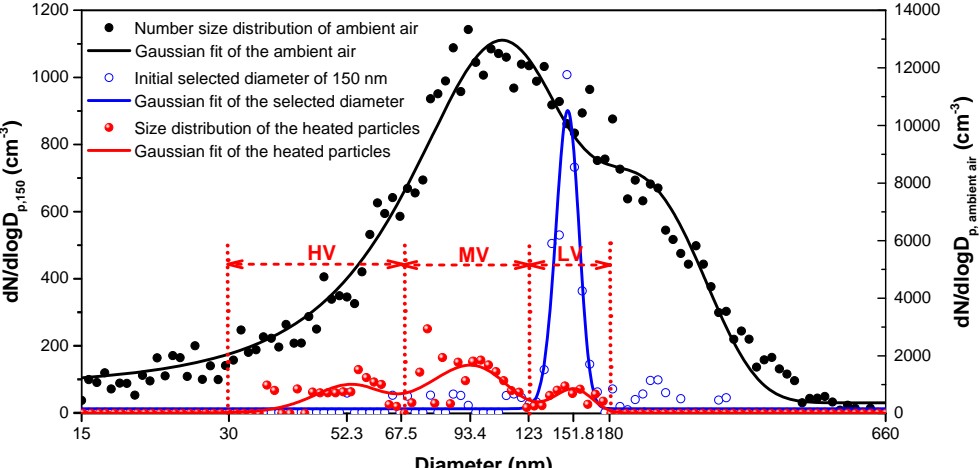

**Figure 2:** Examples of the particle number size distributions of ambient aerosol (black points), as well as the VTDMA measured room temperature bypass sample (~ 25 °C, blue open circles), and the sample passed through a custom-built heating tube at 300 °C ($D_{p, 300\,°C}$, red points) for the initial selected diameter of 150 nm ($D_p$). The corresponding Gaussian fit of the size distributions are shown as black, blue, and red lines, respectively. The size distribution obtained after heating was divided into three size ranges according to previously reported empirical cutting diameters : (1) Particles with diameters $D_{p, 300\,°C}/D_p < 45\%$ were denoted as "high-volatility" (HV), and were not considered as BC. (2) Particles with diameters $45\% < D_{p, 300\,°C}/D_p < 82\%$ were considered as internally mixed BC particles (a nonvolatile core coated with a volatile shell), and were denoted as "medium-volatility" (MV). (3) Particles with diameters $82\% < D_{p, 300\,°C}/D_p < 120\%$ were denoted as "low-volatility" (LV), and were considered as externally mixed BC.



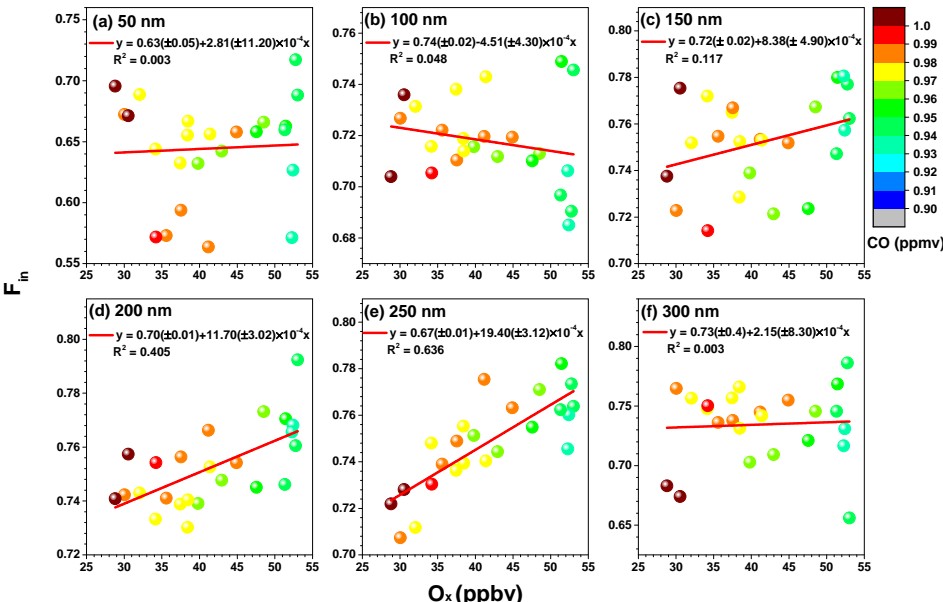

**Figure 3:** Correlation between the diurnally-averaged number fractions of internally mixed BC ($F_{in}$) and the photochemical oxidant ($O_x$) mixing ratios for different size bins (50, 100, 150, 200, 250, and 300 nm). Data points are color-coded with respect to the concentrations of
10 CO (an indicator of primary BC emissions). Low $F_{in}$ values generally appear at high CO concentrations, and vice versa. For 150, 200, and 250 nm diameters, $F_{in}$ values increased with oxidant concentration. The slope of the linear regression (red line) is representative of the oxidation rate of $F_{in}$ (the fit standard error is shown in brackets).



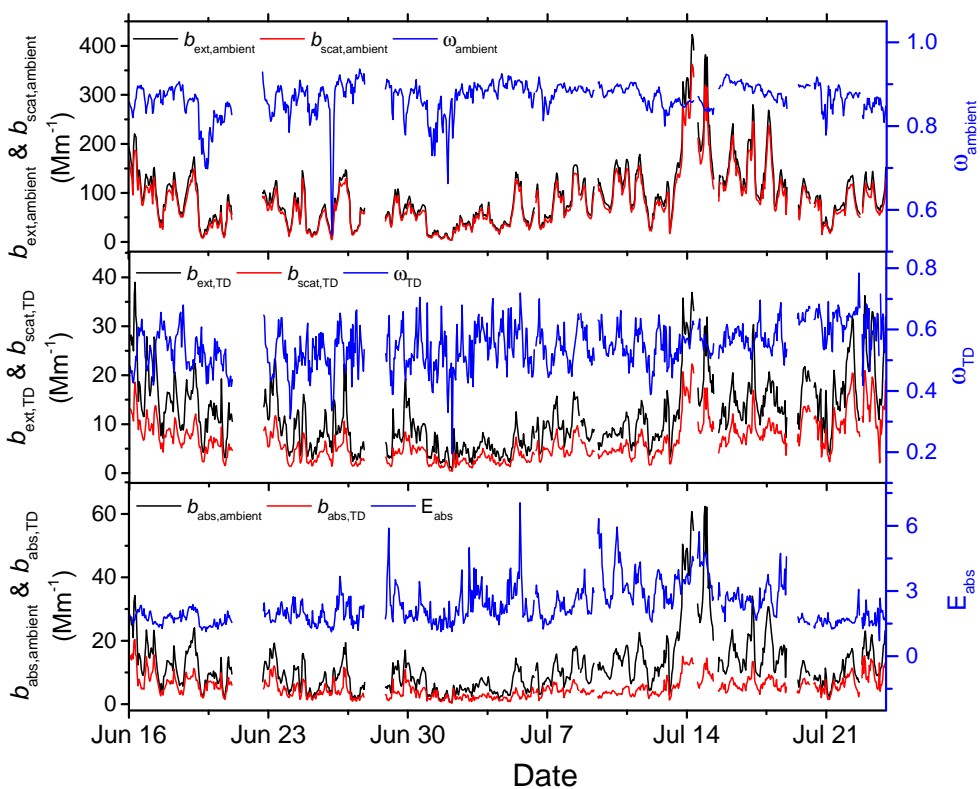

**Figure 4:** Time series of the optical parameters and absorption enhancement (E$_{abs}$) at λ = 532 nm at a time resolution of 10 min. Properties shown are the extinction ($b_{ext}$), scattering ($b_{scat}$), and absorption coefficients ($b_{abs}$), the SSA (ω) of ambient particles ($b_{ambient}$) and particles passed though the thermodenuder ($b_{TD}$) at 300 ºC (after correcting for particle losses).




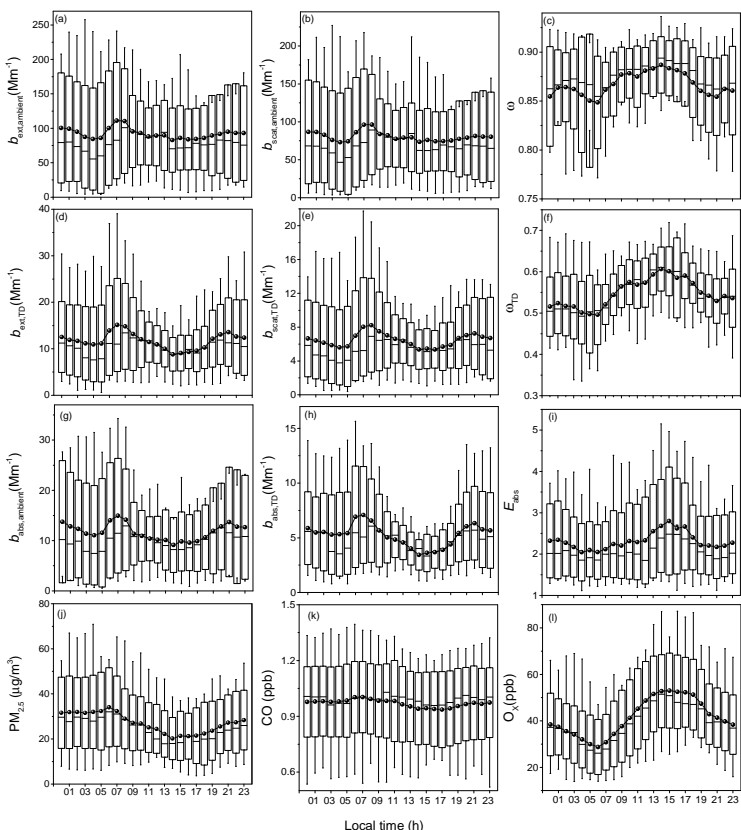

**Figure 5:** The observed diurnal variation of aerosol optical parameters (extinction ($b_{ext}$), scattering ($b_{scatt}$), absorption ($b_{abs}$) coefficients, and SSA ($\omega$)) of ambient particles ($b_{ambient}$, (a)-(c), (g)) and particles passed through the thermodenuder at 300 ºC after correcting for particle losses ($b_{TD}$, (d)-(f), (h)). The absorption enhancement ($E_{abs}$, (i)) was calculated as the ratio between $b_{abs,\,ambient}$ /$b_{abs,\,TD}$. The mass concentrations of $PM_{2.5}$ (j), and the mixing ratios of CO (k) and $O_x$ (l) are also shown for assessing the effect of photochemical oxidation. The optical measurement at $\lambda$ = 532 nm covered the period June 16 to July 23 2016. The box and whisker plots show the mean (dots), median (center solid line), lower and upper quartile (boxes) and 5[th] and 95[st] percentile (whiskers).



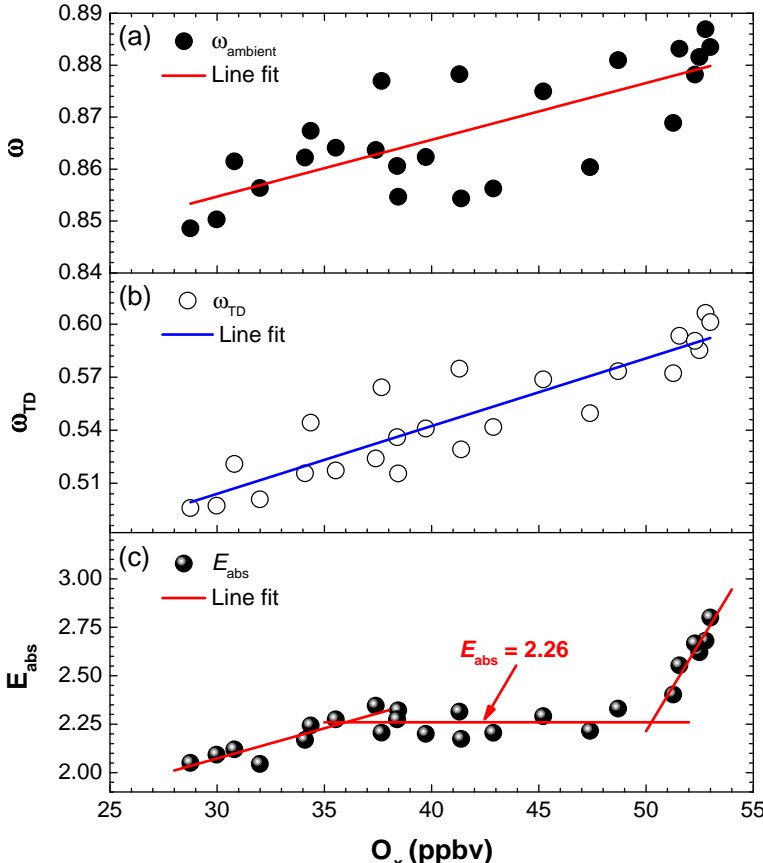

**Figure 6:** Relationship between diurnally-averaged (a) ω, (b) $\omega_{TD}$, and (c) $E_{abs}$ with diurnally-averaged $O_x$ concentrations. A linear orthogonal distance regression fit of the data is shown in the figure for better representing the trends of corresponding parameters with the increment of the oxidant concentration.





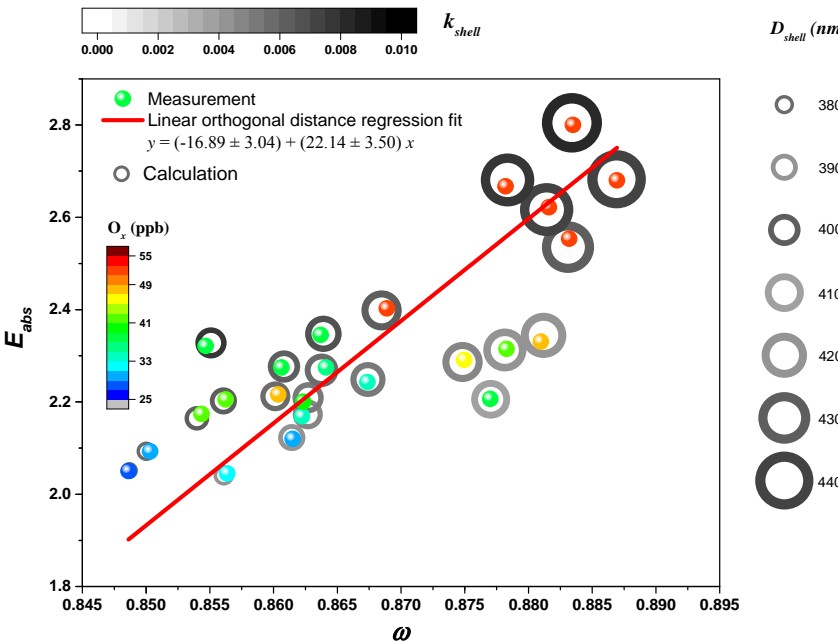

**Figure 7:** Scatter plot of $E_{abs}$ and ω for different photochemical oxidant concentrations. Variation of the observed diurnally-averaged
10   absorption enhancement and SSA (solid points, color-coded with respect to the concentrations of $O_x$) is used for the linear fit. Both $E_{abs}$
and ω increase with $O_x$ mixing ratio. The open circles are the single-particle Mie theory calculation results with an optimized BC core size
of 160 nm. The CRI of BC was fixed at $1.85 + i\ 0.71$. The real part of the CRI of the coating material was fixed at 1.55. The changes of the
imaginary part of the CRI and the thickness of the coating material were color-coded and shown as the different dimensions open circle.





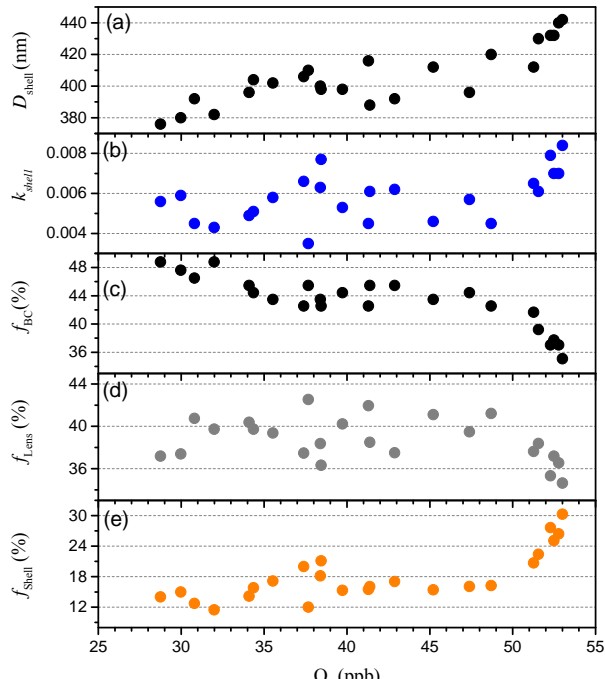

**Figure 8:** Relationship between the calculated values of : (a) thickness ($D_{Shell}$) and (b) imaginary part of the CRI ($k_{Shell}$) of the coated
10   materials, and the relative contribution of (c) the absorption of BC ($f_{BC}$), (d) lensing effect ($f_{Lens}$), and (e) absorption of the shell ($f_{Shell}$) with
$O_x$ concentrations.