# Peer review of "Influence of photochemical aging on light absorption of atmospheric black carbon and aerosol single scattering albedo"

_Atmospheric Chemistry and Physics, 2018_

## Short Comment (SC1) · 21 May 2018

The authors conducted direct field measurement of size-resolved mixing state, absorption enhancement, and single scattering albedo for BC particles during photochemical aging. It could help to advance the current understanding in the large variation of BC absorption during atmospheric aging processes. I have one short comment on the uncertainty associated with the calculation of BC optical properties.

The computation of BC optical properties in this study assumed the core-shell coating structure. However, more and more observations (e.g., China et al., 2015; Wang et al., 2017) have shown various BC coating structures/morphology during aging processes,

which are not core-shell. Further modeling studies (e.g., Scarnato et al., 2013; He et al., 2015, 2016) have indicated a large variation in BC optical properties due to the observed complex coating morphology. Thus, assuming a core-shell structure may lead to uncertainty in the estimate of BC optical properties. Besides, He et al. (2015) also proposed a BC optics-aging mechanism with three evolutional aging stages, which may be useful for the authors' analysis. I suggest that the authors include these recent studies and add some discussions on this important issue.

References

China, S., et al.: Morphology and mixing state of aged soot particles at a remote marine free troposphere site: Implications for optical properties, Geophys. Res. Lett., 42, 1243–1250, doi:10.1002/2014gl062404, 2015.

He, C., et al.: Variation of the radiative properties during black carbon aging: theoretical and experimental intercomparison, Atmos. Chem. Phys., 15, 11967-11980, doi:10.5194/acp-15-11967-2015, 2015.

He, C., et al.: Intercomparison of the GOS approach, superposition T-matrix method, and laboratory measurements for black carbon optical properties during aging, J. Quant. Spectrosc. Radiat. Transf., 184, 287–296, doi:10.1016/j.jqsrt.2016.08.004, 2016.

Scarnato, B. V., et al.: Effects of internal mixing and aggregate morphology on optical properties of black carbon using a discrete dipole approximation model, Atmos. Chem. Phys., 13, 5089–5101, doi:10.5194/acp-13-5089-2013, 2013.

Wang, Y., et al.: Fractal dimensions and mixing structures of soot particles during atmospheric processing, Environ. Sci. Technol. Lett., 4, 487-493, doi:10.1021/acs.estlett.7b00418, 2017.

---

## Referee Comment (RC1) · Anonymous Referee #1 · 26 May 2018

This paper is well written and has considered many aspects in terms of absorption enhancement of BC, but it will be more convincing after addressing the following points:

The Esca vs Ox and SSA vs Ox plots seem crucial, however it seems only overall hourly mean values (from the diurnal variation of entire experimental period) was used, I would say maybe making the scattering plot for all of the data points (maybe hourly average), then bin it in Ox, for each bin, giving the mean/median/percentile etc. you need to make this plot solid, also give the fitting function in the plot.

It would be useful to point out Ox actually may just determine how much secondary aerosol is formed, i.e. increasing the overall ensemble of PM, then increase SSA.

[Figure]

There may not be too much exciting to see the Ox positively correlated with SSA.

The collapse concept is repeatedly discussed but there is no support in your work, how could you say the flat Eabs (even may not be flat after you put all data points in) in the medium Ox is compact soot or not? Could you somehow prove the collapse you are "guessing"? if you can't prove, it is not necessary to emphasize this at many places but just report the solid results you have.

What is the reason to plot $\omega$TD vs Ox? That means some of the low-volatile coating has not been removed, then you will underestimate the Eabs? (it has been mentioned in the text but would be good if this could be properly included)

For your calculation at section 3, could you point out the uncertainty you have by assuming the fixed core size? But I presume you need to use a size distribution of core size? And how did you apply the coated size distribution upon core size?

I am still struggling to understand what is the point for section 4, your results on the Eabs only represent a single ground measurement with limited sources, not even open biomass burning etc. how could be recommend for global models. Also, the surface measurement cannot necessarily represent the columnar information. I would hesitate to expand your work that big given you haven't really done this job.
* * *

---

## Referee Comment (RC2) · Anonymous Referee #2 · 3 Jun 2018

This study investigates the size-resolved mixing state, absorption enhancement (Eabs) and single scattering albedo (SSA) of ambient aerosol at a rural site in East China. They found diurnal variability of Eabs, SSA and a proxy of photochemical aging (Ox). The authors suggested a three stage Eabs process at difference degree of atmospheric photochemical aging. Absorption enhancement is an important topic and more field measurements are needed to understand the variability of Eabs at different geographical locations and under different pollutant conditions. Overall, the manuscript is well written. The paper is worth publishing, but some of the points need to be explained. Authors suggest that the three stage Eabs process is due to collapsed semispherical to highly compact spherical morphology of BC without any morphological data. Authors

should be careful to make this claim. Why the Eabs was stable for Ox mixing ratio between 35 and 50 ppbv? Some of the figures (figs. 7 and 8) need expanded discussion. Authors found relatively higher Eabs compared to other study at similar wavelength. Authors should add some discussion on this.

Specific comments:

Please provide details of the detection limit of the scattering and extinction measurement. Also explain how the uncertainties in extinction, scattering and absorption coefficients were estimated.

Explain how the optical loss of the thermodenuded aerosol were estimated and also contribution of different uncertainties in Eabs measurements.

Authors suggested increase in SSA-TD may be due to incomplete volatilization of non-volatile matter and generation of LV-oxygenated organic aerosol. It will be interesting to see how sca and abs response to TD at different Ox.

Authors found average Eabs of 2.3 at 532 nm which is even higher than Lack et (2012) for forest fire samples (1.4 at 532 nm). Normally BC particles from forest fire are heavily coated and one would expect high Eabs. However, it is not clear if airmasses investigated in this study were also influenced by biomass burning? Or Authors suggest that at higher degree of photochemical aging one would expect higher Eabs compared to thickly coated BC from forest fire?
* * *

---

## Author Comment (AC1) · 7 Jul 2018

**Response to He's comments**

*The authors conducted direct field measurement of size-resolved mixing state, absorption enhancement, and single scattering albedo for BC particles during photochemical aging. It could help to advance the current understanding in the large variation of BC absorption during atmospheric aging processes. I have one short comment on the uncertainty associated with the calculation of BC optical properties.*

Thank you for your helpful comments. Point-by-point responses to the comments are attached below.

*The computation of BC optical properties in this study assumed the core-shell coating structure. However, more and more observations (e.g., China et al., 2015; Wang et al., 2017) have shown various BC coating structures/morphology during aging processes, which are not core-shell. Further modeling studies (e.g., Scarnato et al., 2013; He et al., 2015, 2016) have indicated a large variation in BC optical properties due to the observed complex coating morphology. Thus, assuming a core-shell structure may lead to uncertainty in the estimate of BC optical properties.*

This is a valuable clarification and limits of our study. Accordingly, we added some discussion of the effect of BC morphology on absorption enhancement in the revised manuscript. The morphologies of BC coating are complex and diverse (China et al., 2015; Wang et al., 2017), which also leads to a large variation in BC optical properties (He et al., 2015; He et al., 2016; Scarnato et al., 2013; Wu et al., 2018). We agree with He that novel models will improve our understanding of BC optical properties, but our study did not have particle size distribution information or morphological information and can therefore not perform this type of complex computation.

Mie theory remains a powerful tool for optical data interpretation (Lack et al., 2012) due to its computational efficiency and applicability to radiative transfer models. Morever, the reliability of the core-shell model has been verified in many optical closure studies (Lack et al., 2012; Ma et al., 2012; S. Liu et al., 2015; Wu et al., 2018). In this study, the core-shell calculation has been demonstrated to be useful to interpret the observation data albeit with the limitations that He points out.

*Besides, He et al. (2015) also proposed a BC optics-aging mechanism with three evolutional aging stages, which may be useful for the authors' analysis. I suggest that the authors include these recent studies and add some discussions on this important issue.*

Done. We added recent studies in the revised manuscript.

  Theoretical and experiment results shown that aging causes the dramatic

changes of BC particle morphology (China et al., 2015; He et al., 2015; He et al., 2016; Scarnato et al., 2013; Wang et al., 2017) and leads to more compact black carbon with higher scattering cross section (Peng et al., 2016; Y. Wu et al., 2018), ...

References:

China, S., Scarnato, B., Owen, R. C., Zhang, B., Ampadu, M. T., Kumar, S., Dzepina, K., Dziobak, M. P., Fialho, P., Perlinger, J. A., Hueber, J., Helmig, D., Mazzoleni, L. R., and Mazzoleni, C.: Morphology and mixing state of aged soot particles at a remote marine free troposphere site: Implications for optical properties, Geophys. Res. Lett., 42, 1243–1250, doi:10.1002/2014gl062404, 2015.

He, C., Liou, K.-N., Takano, Y., Zhang, R., Levy Zamora, M., Yang, P., Li, Q., and Leung, L. R.: Variation of the radiative properties during black carbon aging: theoretical and experimental intercomparison, Atmos. Chem. Phys., 15, 11967-11980, doi:10.5194/acp-15-11967-2015, 2015.

He, C., Takano, Y., Liou, K.-N., Yang, P., Li, Q., and Mackowski, D. W.: Intercomparison of the GOS approach, superposition T matrix method, and laboratory measurements for black carbon optical properties during aging, J J. Quant. Spectrosc. Radiat. Transf., 184, 287–296, doi:10.1016/j.jqsrt.2016.08.004, 2016.

Scarnato, B. V., Vahidinia, S., Richard, D. T., and Kirchstetter, T. W.: Effects of internal mixing and aggregate morphology on optical properties of black carbon using a discrete dipole approximation model, Atmos. Chem. Phys., 13, 5089–5101, doi:10.5194/acp-13-5089-2013, 2013.

Wang, Y., Liu, F., He, C., Bi, L., Cheng, T., Wang, Z., Zhang, H., Zhang, X., Shi, Z., and Li, W.: Fractal dimensions and mixing structures of soot particles during atmospheric processing, Environ. Sci. Tech. Let., 4, 487–493, doi:10.1021/acs.estlett.7b00418, 2017.

Wu, Y., Cheng, T., Liu, D., Allan, J. D., Zheng, L., and Chen, H.: Light absorption enhancement of black carbon constrained by particle morphology, Environ. Sci. Technol, 52, 6912–6919, doi:10.1021/acs.est.8b00636, 2018.

*References*

*China, S., et al.: Morphology and mixing state of aged soot particles at a remote marine free troposphere site: Implications for optical properties, Geophys. Res. Lett., 42, 1243–1250, doi:10.1002/2014gl062404, 2015.*

*He, C., et al.: Variation of the radiative properties during black carbon aging: theoretical and experimental intercomparison, Atmos. Chem. Phys., 15, 11967-11980,*

*doi:10.5194/acp-15-11967-2015, 2015.*

*He, C., et al.: Intercomparison of the GOS approach, superposition T-matrix method, and laboratory measurements for black carbon optical properties during aging, J. Quant. Spectrosc. Radiat. Transf., 184, 287–296, doi:10.1016/j.jqsrt.2016.08.004, 2016.*

*Scarnato, B. V., et al.: Effects of internal mixing and aggregate morphology on optical properties of black carbon using a discrete dipole approximation model, Atmos. Chem. Phys., 13, 5089–5101, doi:10.5194/acp-13-5089-2013, 2013.*

*Wang, Y., et al.: Fractal dimensions and mixing structures of soot particles during atmospheric processing, Environ. Sci. Technol. Lett., 4, 487-493, doi:10.1021/acs.estlett.7b00418, 2017.*

---

## Author Comment (AC2) · 7 Jul 2018

**Response to Reviewer 1**

*This paper is well written and has considered many aspects in terms of absorption enhancement of BC, but it will be more convincing after addressing the following points:*

We thank the reviewer for his or her thoughtful and thorough reviews. Point-by-point responses to the comments are attached below. We have made corresponding modifications/revisions based on the inputs, and these changes are marked in the revised manuscript.

*The $E_{abs}$ vs $O_x$ and SSA vs $O_x$ plots seem crucial, however it seems only overall hourly mean values (from the diurnal variation of entire experimental period) was used, I would say maybe making the scattering plot for all of the data points (maybe hourly average), then bin it in $O_x$, for each bin, giving the mean/median/percentile etc. you need to make this plot solid, also give the fitting function in the plot.*

Diurnal variations in atmospheric chemistry would be influenced by underlying daily patterns in emissions, oxidation chemistry, and meteorological variables, all of which could plausibly influence both $E_{abs}$ and $O_x$. Such influences could obscure values binned by $O_x$ averages. We therefore believe diurnal averages are the most appropriate way to treat the influence of atmospheric photochemical aging on the mixing state and optical properties of BC containing particles.

We note that in gas-phase radical chemistry (Monks, 2005) and photochemical oxidation chemistry (Liu et al., 2012; Whalley et al., 2018), diurnal variation is used for modeling analysis because it reduces the influence caused by day to day variations in weather conditions and provides a clearer trend of the observed parameter.

References:

Liu, Z., Wang, Y., Gu, D., Zhao, C., Huey, L. G., Stickel, R., Liao, J., Shao, M., Zhu, T., Zeng, L., Amoroso, A., Costabile, F., Chang, C.-C., and Liu, S.-C.: Summertime photochemistry during CAREBeijing-2007: ROx budgets and $O_3$ formation, Atmos. Chem. Phys., 12, 7737-7752, https://doi.org/10.5194/acp-12-7737-2012, 2012.

Monks, P. S.: Gas-phase radical chemistry in the troposphere, Chem. Soc. Rev., 34, 376-395, 2005.

Whalley, L. K., Stone, D., Dunmore, R., Hamilton, J., Hopkins, J. R., Lee, J. D., Lewis, A. C., Williams, P., Kleffmann, J., Laufs, S., Woodward-Massey, R., and Heard, D. E.: Understanding in situ ozone production in the summertime through radical observations and modelling studies during the Clean air for

London project (ClearfLo), Atmos. Chem. Phys., 18, 2547-2571, https://doi.org/10.5194/acp-18-2547-2018, 2018.

*It would be useful to point out $O_x$ actually may just determine how much secondary aerosol is formed, i.e. increasing the overall ensemble of PM, then increase SSA. There may not be too much exciting to see the $O_x$ positively correlated with SSA.*

Done. We added the clarification in Section 3.3 in the revised manuscript.

Positive correlations between ω, $ω_{TD}$ and $O_x$ concentrations were observed in our measurements (Fig. 6 (a) and (b)), which suggests that higher $O_x$ actually increases the mass fraction of secondary aerosol particles and the overall ensemble of particle material and SSA.

*The collapse concept is repeatedly discussed but there is no support in your work, how could you say the flat $E_{abs}$ (even may not be flat after you put all data points in) in the medium $O_x$ is compact soot or not? Could you somehow prove the collapse you are "guessing"? if you can't prove, it is not necessary to emphasize this at many places but just report the solid results you have.*

Done. Since we do not have microscopic images to support our speculation, we removed the emphasis on the collapse concept in the revised manuscript.

*What is the reason to plot $ω_{TD}$ vs $O_x$? That means some of the low-volatile coating has not been removed, then you will underestimate the $E_{abs}$? (it has been mentioned in the text but would be good if this could be properly included)*

Liu et al. (2015) report on the comparison of $E_{abs}$ measured with two different methods: TD operating at 250 °C and MAE (mass absorption efficiency) method. They found that $E_{abs}$ values with these two methods agreed closely, which indicated that the non- and low-volatile coating did not have a notable impact on $E_{abs}$'s measurement. In this work, the TD was operated at 300 °C and the low-volatile coating should not affect the measurement.

Our reported diurnal hourly average $ω_{TD}$ ranged from 0.50 to 0.61, which is comparable with the range (0.50 - 0.60, λ = 405 nm) reported by Radney et al. (2014) for laboratory-generated soot aerosol. In Fig. 6(b), we can see that $ω_{TD}$ increased as the increasing of the $O_x$ concentration, which is mostly due to the morphology change during photooxidation aging (China et al., 2015). Recent results reported by Peng et al. (2016) and Y. Wu et al. (2018) show that aging causes dramatic changes of BC particle morphology and leads to more compact black carbon. As demonstrated by Radney et al. (2014) and Forestier et al. (2018), particle collapse leads to an increase

of ω. In this regard, the plot of $\omega_{TD}$ *vs* $O_x$ can be used as an indicator for the changes of BC morphology.

We added following discussion in the revised manuscript.

[revised manuscript text omitted]

*For your calculation at section 3, could you point out the uncertainty you have by assuming the fixed core size? But I presume you need to use a size distribution of core size? And how did you apply the coated size distribution upon core size?*

Since we did not have particle size distribution information in this study, we assumed monodisperse particles in the Mie calculation. This method follows that demonstrated by Saleh et al. (2015).

We added the following discussion in the revised manuscript.

In this work, the particle size distribution information was not available. A method based on single-particle core-shell Mie theory (Bohren and Huffman, 1983; Saleh et al., 2015) was developed to interpret the proposed three-stage

aging mechanism observed in this work. The sensitivity of this assumption is discussed in Sect. S6 in the supplement.

**Reference:**

Saleh, R., Marks, M., Heo, J., Adams, P. J., Donahue, N. M., and Robinson, A. L.: Contribution of brown carbon and lensing to the direct radiative effect of carbonaceous aerosols from biomass and biofuel burning emissions, J. Geophys. Res. Atmos. 120, 10285–10296, doi:10.1002/2015JD023697, 2015.

We added following discussion in the Sect. S6 in the supplement.

[Figure]

**Figure S15:** Comparison of the Mie theory results of $E_{abs}$ with monodisperse BC core with 160 nm diameter and polydisperse size distributions with a geometric standard deviation of 1.6 and mode diameters of 160, 120, 100, and 80 nm, respectively. The parameters used for the calculation are the same as in Fig. 7. Polydisperse BC core sizes have larger $E_{abs}$ values than monodisperse BC core; however, the trends of $E_{abs}$ values are same. We use an optimization process for determining $D_{core}$, $D_{shell}$, and $k_{shell}$; similar trends show that monodisperse BC core can be used for the interpretation of the measurement data.

[Figure]

**Figure S16:** Fractional contribution of the lensing effect ($f_{Lens}$), the absorption of BC ($f_{BC}$) and the shell ($f_{Shell}$) to absorption enhancement with different BC core sizes as Fig. S15. The trends of the diurnal pattern are similar. The differences between these calculations are less than 10%.

*I am still struggling to understand what is the point for section 4, your results on the $E_{abs}$ only represent a single ground measurement with limited sources, not even open biomass burning etc. how could be recommend for global models. Also, the surface measurement cannot necessarily represent the columnar information. I would hesitate to expand your work that big given you haven't really done this job.*

Done. We removed this section from the revised manuscript.

---

## Author Comment (AC3) · 7 Jul 2018

**Response to Reviewer 2**

*This study investigates the size-resolved mixing state, absorption enhancement ($E_{abs}$) and single scattering albedo (SSA) of ambient aerosol at a rural site in East China. They found diurnal variability of $E_{abs}$, SSA and a proxy of photochemical aging ($O_x$). The authors suggested a three stage $E_{abs}$ process at difference degree of atmospheric photochemical aging. Absorption enhancement is an important topic and more field measurements are needed to understand the variability of $E_{abs}$ at different geographical locations and under different pollutant conditions. Overall, the manuscript is well written. The paper is worth publishing, but some of the points need to be explained.*

We thank the reviewer for his or her thoughtful and thorough reviews. Point-by-point responses to the comments are attached below. We have made corresponding modifications/revisions based on the inputs, and these changes are marked in the revised manuscript.

*Authors suggest that the three stage $E_{abs}$ process is due to collapsed semispherical to highly compact spherical morphology of BC without any morphological data. Authors should be careful to make this claim.*

Done. Recent studies (China et al., 2015; He et al., 2015; Peng et al., 2016; Wu et al., 2018) on the morphology shown that aging causes dramatic changes of BC particle morphology and leads to more compact black carbon. Our speculation on the collapse concept is reasonable. However, according to the suggestion of Reviewer 1, since we did not have morphological data, we removed the emphasis on the collapse concept in the revised manuscript. We just report the solid results we have.

References:

China, S., Scarnato, B., Owen, R. C., Zhang, B., Ampadu, M. T., Kumar, S., Dzepina, K., Dziobak, M. P., Fialho, P., Perlinger, J. A., Hueber, J., Helmig, D., Mazzoleni, L. R., and Mazzoleni, C.: Morphology and mixing state of aged soot particles at a remote marine free troposphere site: Implications for optical properties, Geophys. Res. Lett., 42, 1243–1250, doi:10.1002/2014gl062404, 2015.

He, C., Liou, K.-N., Takano, Y., Zhang, R., Levy Zamora, M., Yang, P., Li, Q., and Leung, L. R.: Variation of the radiative properties during black carbon aging: theoretical and experimental intercomparison, Atmos. Chem. Phys., 15, 11967-11980, doi:10.5194/acp-15-11967-2015, 2015.

Peng, J., Hua, M., Guo, S., Du, Z., Zheng, J., Shang, D., Zamora, M. L., Zeng, L., Shao, M., Wu, Y. S., Zheng, J., Wang, Y., Glen, C. R., Collins, D. R., Molina, M. J., and Zhang, R.: Markedly enhanced absorption and direct radiative forcing of black carbon under polluted urban environments, Proc. Natl. Acad. Sci. U S A, 113, 4266–4271, doi:10.1073/pnas.1602310113, 2016.

Wu, Y., Cheng, T., Liu, D., Allan, J. D., Zheng, L., and Chen, H.: Light absorption enhancement of black carbon constrained by particle morphology, Environ. Sci. Technol, 52, 6912–6919, doi:10.1021/acs.est.8b00636, 2018.

*Why the $E_{abs}$ was stable for $O_x$ mixing ratio between 35 and 50 ppbv? Some of the figures (figs. 7 and 8) need expanded discussion. Authors found relatively higher $E_{abs}$ compared to other study at similar wavelength. Authors should add some discussion on this.*

Recent morphologically constrained modelling developed by Y. Wu et al. (2018) demonstrated that after full aging, the BC particles became a more compact aggregation, which leads to a stable range of $E_{abs}$ (averaged value ~ 2.5, with a minimum value of ~ 2 and a maximum value of ~ 3.5). In this study, the observed stable $E_{abs}$ value ($2.26 \pm 0.06$) for $O_x$ mixing ratios between 35 and 50 ppbv was in the range of Y. Wu et al.'s result, suggesting that BC was fully aged under this oxidation level. In addition, the calculated values of $D_{shell}$ and $k_{shell}$ in Fig. 8 were fairly constant in the stable stage, which means a stable contribution of the non-BC components. Fully aged BC and minor changes of non-BC components lead to a stable $E_{abs}$ for $O_x$ mixing ratio between 35 to 50 ppbv. The $E_{abs}$ value reported in our work was relatively higher than others, but was still reasonable.

We added following discussion in Sect. 3.3 in the revised manuscript.

Recent morphologically constrained modelling developed by Y. Wu et al. (2018) demonstrated that after full aging the BC particles became a more compact aggregation, which leads to a stable range of $E_{abs}$ (averaged value ~ 2.5, with a minimum value of ~ 2 and a maximum value of ~ 3.5). Our results fall within this range and suggest that BC was fully aged under this oxidation level.

We added the following discussion for Fig. 7.

The solid points are the observed results and color-coded with respect to the concentrations of $O_x$. The open circles are the single-particle Mie core-shell modeled results with an optimized BC core size of 160 nm, and color- and size-coded with respect to the imaginary part of the CRI of coating material ($k_{shell}$) and the diameter of coating material ($D_{shell}$), respectively. The color-coded plot shows the connection between $E_{abs}$, SSA and atmospheric

photochemistry. The modeled results are consistent with the observed results. Both SSA and $E_{abs}$ values rise with increasing $D_{shell}$ and $k_{shell}$, indicating that the coating thickness and absorption play key roles in determining SSA and $E_{abs}$.

We added following discussion for Fig. 8.

The fractional contribution of $f_{BC}$, $f_{Lens}$, $f_{Shell}$ ranged from 35-49%, 35-42%, and 11-30%, respectively, with a mean value of 43±4%, 39±2%, and 18±5%. A ternary plot is shown in the supplement Fig. S14. At the first stage of Fig. 6(c), $D_{shell}$ increased with $O_x$ concentrations, but $k_{shell}$ showed an obscure variation with increasing $O_x$ mixing ratios. The rise in $E_{abs}$ was mainly caused by the thicker coating. In the second stage, with constant $E_{abs}$, all the parameters ($D_{shell}$, $k_{shell}$, $f_{BC}$, $f_{Lens}$, and $f_{Shell}$) remained fairly constant, which suggests a stable contribution of the non-BC components. Compact aggregation of fully aged BC and minor changes of non-BC coating materials lead a stable $E_{abs}$ for $O_x$ mixing ratios between 35 and 50 ppbv.

References:

Wu, Y., Cheng, T., Liu, D., Allan, J. D., Zheng, L., and Chen, H.: Light absorption enhancement of black carbon constrained by particle morphology, Environ. Sci. Technol., 52, 6912–6919, doi:10.1021/acs.est.8b00636, 2018.

*Specific comments:*
*Please provide details of the detection limit of the scattering and extinction measurement. Also explain how the uncertainties in extinction, scattering and absorption coefficients were estimated.*

The details of the detection limit and uncertainties of the instrument have been described in our previously published papers (Zhao et al., 2014; Xu et al., 2016; Fang et al., 2017). In this paper, we just give a brief introduction.

We added following explanation in the revised manuscript.

The details of the evaluation of the instrument have been described in our previously published paper (Zhao et al., 2014; Xu et al., 2016; Fang et al., 2017). Detection limits of each parameter were determined by using an Allan variance analysis.

The total uncertainties (summed in quadrature of each error source) in extinction, scattering, absorption coefficients, and SSA measurements were estimated to be less than 4%, 3%, 5%, and 4%, respectively.

Reference:

Fang, B., Zhao, W., Xu, X., Zhou, J., Ma, X., Wang, S., Zhang, W., Venables, D.S., and Chen, W.: Portable broadband cavity-enhanced spectrometer utilizing Kalman filtering: application to real-time, in situ monitoring of glyoxal and nitrogen dioxide, Opt. Express, 25, 26910-26922, doi:10.1364/OE.25.026910, 2017.

Xu, X., Zhao, W., Zhang, Q., Wang, S., Fang, B., Chen, W., and Gao, X.: Optical properties of atmospheric fine particles near Beijing during the HOPE-J3A campaign, Atmos. Chem. Phys., 16, 6421-6439, doi:10.5194/acp-16-6421-2016, 2016.

Zhao, W., Xu, X., Dong, M., Chen, W., Gu, X., Hu, C., Huang, Y., Gao, X., Huang, W., and Zhang, W.: Development of a cavity-enhanced aerosol albedometer, Atmos. Meas. Tech., 7, 2551-2566, doi:10.5194/amt-7-2551-2014, 2014.

*Explain how the optical loss of the thermodenuded aerosol were estimated and also contribution of different uncertainties in $E_{abs}$ measurements.*

The optical loss of the TD was explained in Sec. S3 in the supplement.

We added following explanation in the revised manuscript.

> The total uncertainty in $E_{abs}$ measurement was ~about 9% (mainly contributed by uncertainties in the measurement of $b_{abs,\ ambient}$ (5%), $b_{abs,\ TD}$ (5%), and particle losses inside TD (6%)).

*Authors suggested increase in SSA-TD may be due to incomplete volatilization of nonvolatile matter and generation of LV-oxygenated organic aerosol. It will be interesting to see how sca and abs response to TD at different Ox.*

The scatter plots of $b_{scat,TD}$, $b_{abs,TD}$, $b_{ext,TD}$, $\omega_{TD}$, and ambient $PM_{2.5}$ concentration versus $O_x$ mixing ratios are shown in Fig. R1. The absorption and scattering coefficient are not intensive optical properties. The values also depended on the sample mass concentration. Very little information can be retrieved from the relationship between $b_{scat,TD}$, $b_{abs,TD}$ and $O_x$ without the mass concentration of the heated sample.

[Figure]

**Figure R1** Scatter plots for (a) $b_{scat,TD}$, $b_{abs,TD}$, $b_{ext,TD}$ and $\omega_{TD}$ versus $O_x$ mixing ratios of the heated sample; (b) ambient $PM_{2.5}$ concentration versus $O_x$ mixing ratios.

*Authors found average $E_{abs}$ of 2.3 at 532 nm which is even higher than Lack et (2012) for forest fire samples (1.4 at 532 nm). Normally BC particles from forest fire are heavily coated and one would expect high $E_{abs}$. However, it is not clear if airmasses investigated in this study were also influenced by biomass burning? Or Authors suggest that at higher degree of photochemical aging one would expect higher $E_{abs}$ compared to thickly coated BC from forest fire?*

Recent studies suggested that the value of $E_{abs}$ not only depends on the coating thickness, but also on the morphology of BC particles (China et al., 2013; 2015; Peng et al., 2016; Wu et al., 2018). $E_{abs}$ value is limited to ~ 5% at the start of the aging, and increases with the collapse and encapsulation of the particle. With the ban of straw burning in China in recent years, outdoor burning of straw is now much less common. The air masses in this study were not affected by biomass burning. Higher $E_{abs}$ value in this study was mainly caused by photochemical aging.

China, S., Mazzoleni, C., Gorkowski, K., Aiken, A. C., and Dubey, M. K.: Morphology and mixing state of individual freshly emitted wildfire carbonaceous particles, Nat. Commun., 4, 2122, doi:10.1038/ncomms3122, 2013.

China, S., Scarnato, B., Owen, R. C., Zhang, B., Ampadu, M. T., Kumar, S., Dzepina, K., Dziobak, M. P., Fialho, P., Perlinger, J. A., Hueber, J., Helmig, D., Mazzoleni, L. R., and Mazzoleni, C.: Morphology and mixing state of aged soot particles at a remote marine free troposphere site: Implications for optical properties, Geophys. Res. Lett., 42, 1243–1250, doi:10.1002/2014gl062404, 2015.

Peng, J., Hua, M., Guo, S., Du, Z., Zheng, J., Shang, D., Zamora, M. L., Zeng, L., Shao, M., Wu, Y. S., Zheng, J., Wang, Y., Glen, C. R., Collins, D. R., Molina, M. J., and Zhang, R.: Markedly enhanced absorption and direct radiative forcing of black carbon under polluted urban environments, Proc. Natl. Acad. Sci. U S A, 113, 4266–4271, doi:10.1073/pnas.1602310113, 2016.

Wu, Y., Cheng, T., Liu, D., Allan, J. D., Zheng, L., and Chen, H.: Light absorption enhancement of black carbon constrained by particle morphology, Environ. Sci. Technol, 52, 6912–6919, doi:10.1021/acs.est.8b00636, 2018.

---

## Author Response (AR2)

**Response to the Reviewer's comments**

We thank the reviewers and editor for their thoughtful and thorough reviews. Point-by-point responses to the comments are attached below.

**Response to editor**
*See reviewers' comments for outstanding issues that require addressing. I would especially draw your attention to the point raised regarding the correlations with $O_x$. Both reviewers are now requesting this and I happen to concur; If there is a relationship between SSA and $E_{abs}$ with $O_x$, then a direct correlation is the best way to test this, rather than a correlation of the diurnal profiles and not showing this in the manuscript undermines the credibility of the manuscript. While examples are given in the rebuttal of this type of graphing, these concern gas phase processes that are more likely to be intrinsically related to actinic flux, which has a very strong diurnal causation. In this instance, if a direct relationship between $O_x$ and the measured parameters doesn't exist, then this would throw into question the conclusions of the paper; it would imply strongly that the correlation noted in diurnal profiles was not causal, i.e. that some other diurnally-modulated processes were responsible for changing the variables in such a way that their average diurnal profiles resembled one another, but they were otherwise unrelated.*

DONE. We perform the hourly scatter plot and bin it in $O_x$ in the following (Fig. R1 to Fig. R3).

Diurnal analysis is not only used for gas phase processes study, but also for studying the main factors governing the diurnal cycles of the aerosol properties (Backman et al., 2012). This work was performed during the summer, when $O_3$ has a central role in the generation of secondary aerosol. The concentration of $O_x$ can be treated as an indicator of photochemical activity in a rural site. Both $O_x$, $PM_1$ and $PM_{2.5}$ concentrations have clear diurnal cycles and similar patterns (Fig. 1 in the manuscript), which means that photochemical is the intrinsically factor that governed the diurnal cycles of the aerosol mass, and closely linked to the optical properties.

    The scattering plot of hourly averaged $E_{abs}$ and $O_x$ is shown in Fig. R1. For statistical analysis, the frequency distributions of $E_{abs}$ and $O_x$ are also shown. The values of the diurnal average data (red dots) have a similar pattern with and were comparable with the mean values (blue dots) of the hourly data that binned in $O_x$ (with a bin size of 5 ppbv). Different from the hourly average of all the data points, diurnal analysis reduces the influence caused by weather condition from day to day

and provides a more precise trend of the observed parameter. The resulting diurnal variation data fell in the region where the frequency distribution is large. Similar results were found for the comparison of ω and $\omega_{TD}$ (as shown in Fig. R2 and Fig. R3).

We therefore believe diurnal averages are the most appropriate way to treat the influence of atmospheric photochemical aging on BC mixing state, and our conclusion with diurnal analysis for this paper is appropriate and creditable.

Reference:

Backman, J., Rizzo, L. V., Hakala, J., Nieminen, T., Manninen, H. E., Morais, F., Aalto, P. P., Siivola, E., Carbone, S., Hillamo, R., Artaxo, P., Virkkula, A., Petäjä, T., and Kulmala, M.: On the diurnal cycle of urban aerosols, black carbon and the occurrence of new particle formation events in springtime São Paulo, Brazil, Atmos. Chem. Phys., 12, 11733-11751, https://doi.org/10.5194/acp-12-11733-2012, 2012.

[Figure]

**Fig. R1** Relationship between $E_{abs}$ and $O_x$ concentrations, and the frequency distribution of the measured $E_{abs}$ and $O_x$. The gray circles are the measurement data with one-hour time resolution. The hourly averaged $E_{abs}$ was then binned in $O_x$ with bin size of 5 ppbv. The corresponding mean (blue solid dot), median (center solid line), lower and upper quartile (boxes) and 10th and 90th percentile (whisker) are shown as the box and whisker plots. The diurnal average data is shown as red solid dot for comparison.

[Figure]

**Fig. R2** Same as Fig. R1, the relationship between ω and $O_x$ concentrations, and the frequency distribution of the measured ω and $O_x$.

[Figure]

**Fig. R3** Same as Fig. R1, the relationship between $\omega_{TD}$ and $O_x$ concentrations, and the frequency distribution of the measured $\omega_{TD}$ and $O_x$.

**Response to Reviewer 1**

*I think the diurnal variation is "ok", though there seemed a very large variation. I doubt the $E_{abs}$ vs $O_x$ variation will still stand or not, which is very crucial for the conclusion. I really want to see simply show a $E_{abs}$ vs $O_x$ plot in hourly average for all of the data points for the whole project, not just those 24 points. Could you please do it.*

DONE. Please see in the reply to the editor.

**Response to Reviewer 2**

*1) Author's response on the observed higher value of $E_{abs}$ compared to previous study is not convincing. I'm still not sure higher $E_{abs}$ reported is mainly due to morphology and not coating. How can authors be sure that the plume is not influenced by biomass burning?*

Both coating and morphology changes will influence the values of $E_{abs}$. Higher $E_{abs}$ values have been demonstrated to be convinced in a recent experimental and theoretical study (Wu et al., 2018, and references therein).

    The observation site is a rural background site surrounded by basic farmland protection areas and has no significant industrial pollution sources or tall buildings nearby. The enforcement of the ban of straw burning in Yangtze River Delta region is very strict in recent years; outdoor burning of the straw cannot be seen during the banned period.

DONE. Please see in the reply to the editor.

[revised manuscript text omitted]

---

## Author Response (AR3)

Comments to the Author:

I thank the authors for generating the requested graphs vs $O_x$, however I deem that these are important enough that they should be included in the manuscript alongside the other analyses and the authors should revise some of their interpretations and conclusions accordingly.

While the diurnal patterns may show a nice relationship, I have yet to see any evidence presented that supports the assumption that this relationship is governed by a single common factor such as photochemistry (see below), so this undermines the logic of this being used as a primary basis for the interpretation. For one, I would note that the $E_{abs}$ vs $O_x$ graph fails to exhibit the uptick in $E_{abs}$ above 50ppbv and would appear to actually decrease above 80ppbv. I take this as clear evidence that the 3-stage model proposed by the authors is unsound and in need of either revision or complete removal from this paper.

Generally, I feel the authors may have misinterpreted my earlier statement concerning diurnal patterns, as the statement made on page 6 of the revised manuscript is factually incorrect. A clear diurnal pattern does not, as stated by the authors, mean that photochemistry is the intrinsic governing factor because the diurnal patterns of urban pollutant concentrations are also very strongly linked to human activity (e.g. traffic, cooking, heating) and boundary layer dynamics. BC will be subject to all of these factors and in many cases the factors are intrinsically difficult to disentangle based on observations alone. For instance, photochemistry is promoted by the increasing actinic flux during the morning, but this also corresponds to an increase in solar heating of the surface, which in turn promotes vertical mixing and the exchange of fresh pollution in the lower parts of the boundary layer with background air from aloft, which may include more aged pollution. Furthermore, there is often a general reduction in primary emissions after the morning rush hour subsides. The net result is that there can potentially be an apparent 'ageing' of the pollutants without having to invoke any photochemistry at all. To explicitly link observed changes in BC properties to in situ photochemistry alone requires all other confounding factors to be discounted, which isn't done here.

Note that this is not to say that photochemistry isn't responsible on some level; if the pollutants with high $E_{abs}$ and $O_x$ concentrations are aged transport from sources upwind, it is still reasonable to assume that photochemistry is ultimately responsible for the ageing; it is not simply in situ photochemistry, which seems to be the working hypothesis here. The authors should rephrase these sections accordingly to treat the underlying processes governing the diurnal profiles with more caution.

Furthermore, I'm not convinced with some of the other responses to the latest round of comments. There is simply no direct evidence presented that the observed changes in $E_{abs}$ are due to morphology rather than coating thickness, so this must be treated as a

speculative explanation unless further evidence can be offered to rule out other effects.

Following from these, there are many inferences drawn that should be addressed before this goes to publication. There is simply no basis for saying that the changes in optical properties are caused by the higher $O_x$ (e.g. Page 9, line 17) because this is merely assuming causation, which I regard as being in doubt. The authors must go through the manuscript and temper their language to this effect; I would recommend that they refer to the increases in $E_{abs}$ being 'associated' with changes in $O_x$, which is a far safer statement.

Overall, I still consider this paper to be publishable because there are some interesting observations and trends noted, however there is currently a tendency to over-interpret the data and make statements that are not supported by the evidence and this must be addressed.

Dear Dr. James Allan,

Thank you very much for your helpful suggestion and your effort to make the manuscript solid. We made corresponding revisions based on your input.

We removed the 3-stage concept, included the scattering plot of hourly data vs. $O_x$ concentration in the supplement, and added necessary discussion in the revised manuscript.

Actually, from the hourly averaged data that binned with $O_x$, if we excluded the data for $O_x$ concentration that larger than 80 ppbv (the corresponding frequency distribution only accounts for less than 1% of the observed data), similar growth process as diurnal data can be seen but with different $O_x$ concentration boundaries for each step, which was probably due to the different ways of data representing.

Thank you for giving us a detailed explanation of the diurnal pattern. We removed the statement on page 6 of the photochemical that governed the diurnal cycles from the revised manuscript and tempered the language about the changes of optical properties associated with the changes in $O_x$.

We checked the presentation of the factor that caused the changes of $E_{abs}$. In our summer time observations, we suggested that 
[revised manuscript text omitted]

(a)

[Figure]

5  (b)

[Figure]

[Figure]

**Figure S10:** Relationship between (a) ω, (b) ωTD, and (c) Eabs with Ox concentrations, and the corresponding frequency distribution of each parameters. The gray circles are the measurement data with one-hour time resolution. The hourly averaged (a) ω, (b) ωTD, and (c) Eabs were then binned in Ox with a bin size of 5 ppbv. The corresponding mean (blue solid dot), median (center solid line), lower and upper quartile (boxes) and 10th and 90th percentile (whisker) are shown as the box and whisker plots. The diurnal average data are shown as red solid dot for comparison.

**S7 Mie theory modelling and attribution of light absorption**

[revised manuscript text omitted]

Authropogenic SOM (Liu et al., 2015)
— Toluene + OH
┈ Toluene + OH + NO$_x$
— $m$-Xylene + OH
┈ $m$-Xylene + OH + NO$_x$
Biogenic SOM (Liu et al., 2013)
— a-pinene + O$_3$
— Limonene + O$_3$
— BrC spheres (Alexander et al., 2008)
— Fulvic acid (Liu et al., 2015)
┈ SRFA (Bluvshtein et al., 2017)

■ Black carbon (Kirchstetter et al., 2004)
■ BB (Kirchstetter et al., 2004)
▼ BB (Chakrabarty et al., 2010)
✖ HULIS (Hoffer et al., 2006)
◁ BB + Authrogenic (Lack et al., 2012)
▲ BB (Lack et al., 2012)
✚ HULIS (Dinar et al., 2008)
● Urban BrC (Cappa et al., 2012)
▶ Fresh BB (J. Sumlin et al., 2017)
◆ Aged BB (J. Sumlin et al., 2017)

⊕ **This Work**

**Figure** S14: Comparison of the retrieved imaginary part of the coated shell with previously reported

5  values of fresh and aged organic material (adapted from P. Liu et al. (2015) and J. Sumlin et al. (2017)).

[Figure]

**Figure S15:** Ternary plot of the fractional contribution of lensing effect ($f_{Lens}$), the absorption of BC ($f_{BC}$) and the shell ($f_{Shell}$) to absorption enhancement (the values of $E_{abs}$ were color-coded).

[Figure]

**Figure S16:** Comparison of the Mie theory results of $E_{abs}$ with monodisperse BC core with 160 nm diameter and polydisperse size distributions with a geometric standard deviation of 1.6 and mode diameters of 160, 120, 100, and 80 nm, respectively. The parameters used for the calculation are the same as in Fig. 7. Polydisperse BC core sizes have larger $E_{abs}$ values than monodisperse BC core; however, the trends of $E_{abs}$ values are same. We use an optimization process for determining $D_{core}$, $D_{shell}$, and $k_{shell}$; similar trends show that monodisperse BC core can be used for the interpretation of the measurement data.

[Figure]

**Figure** S17: Fractional contribution of the lensing effect ($f_{Lens}$), the absorption of BC ($f_{BC}$) and the shell ($f_{Shell}$) to absorption enhancement with different BC core sizes as Fig. S15. The trends of the diurnal pattern are similar. The differences between these calculations are less than 10%.

---

## Author Response (AR4)

**Response to Editor's comments**

*Comments to the Author: I am very concerned that the key issue identified by the reviewers and myself is not being addressed to the point that it is being consciously avoided. Specifically, I asked that the analysis vs $O_x$ be included in the manuscript, but instead, the authors have merely included this in the supplement, while leaving the original graphs in place in the main manuscript. Furthermore, while the 3-stage model has been taken out (although the three fit lines are still present in figure 6), the authors still discuss a 2-stage model on page 10 as follows, "In the second region, $E_{abs}$ was unchanged (2.26 ± 0.06) for $O_x$ mixing ratios between 35 and 50 ppbv. These two regions are most likely corresponded to Peng et al.'s (2016) two-stage morphology variation mechanism, in which collapsed semispherical BC is transformed to a spherical morphology (Gustafsson and Ramanathan, 2016)." This statement is in contradiction to $E_{abs}$ analysis binned according to $O_x$ on figure S10, which shows a more monotonic increase in $E_{abs}$ up to about 45 ppbv. Therefore, for the statement on page 10 to stand, the diurnally-averaged method must have to be shown to be more statistically or scientifically valid than the $O_x$-binned method.*

*In choosing to ignore the $O_x$ binned data in favour of the diurnally-binned data, the clear implication is that the authors consider the diurnal method to be the more scientifically meaningful, in spite of the fact that the reviewers consider the $O_x$-binned method to be a more obvious approach. The onus is therefore on the authors to present a robust case for the validity of the diurnal method, but unfortunately, I don't see this in the current manuscript. The justification is given as follows: "Different from the hourly average of all the data points, diurnal analysis reduces the influence caused by weather condition from day to day and provides a more smooth trend of the observed parameter, which has been demonstrated in the study of the main factors governing the diurnal cycles of the aerosol properties (Backman et al., 2012)." This justification I regard as being wholly insufficient. To take it point by point:*

*1. The 'weather conditions' referred to will include sunlight, so in reducing this influence, this averaging method will accordingly diminish the role of photochemistry relative to activity-based emissions patterns (which will not be diminished), thereby making this analysis less, not more, valid to the working hypothesis.*
*2. Stating that the data is 'more smooth' is not a complete justification because one also needs to demonstrate that the smoothness is a result of the data being more meaningful, not that the data has been artificially smoothed on a mathematical level, which can introduce erroneous artefacts.*
*3. The Backman et al. reference does not provide a precedent for the methodology used here. While this paper does present diurnal averages, it does not scatter plot diurnally averaged data, as has been done here.*

*I had originally hoped that the inclusion of this $O_x$-binned analysis would be relatively straightforward in this paper as the results are qualitatively the same, so on*

*the last version, I had marked the revisions as 'minor'. However on seeing this version of the paper, I am now concerned that there is too much of the work that is invested in the quantitative comparison of data after diurnal averaging, which I am yet to be convinced is a sound approach. As such, I have decided to mark this version down as needing 'major' revisions. In order for me to consider this publishable, I would consider that the authors must do one of the following:*

*1. Present a more robust justification of the diurnal methodology (which I am now becoming increasingly concerned is not possible)*
*2. More clearly compare the diurnal method with other binning approaches (i.e. in the main manuscript) and discuss the similarities and differences, and restrict the interpretation of the diurnal comparisons to purely qualitative, rather than quantitative, analysis.*
*3. Redo the quantitative analysis using a more scientifically defensible binning approach.*

Dear Dr. James Allan,

Thank you very much for your helpful comments and your effort to make the manuscript solid. We made corresponding revisions based on your suggestions.

1) We completely removed the 3-stage concept in the revised manuscript.

2) Due to the lack of the supporting data of "weather conditions" to distinguish the contribution of the photochemistry to the diurnal patterns, the scattering plot of two diurnally averaged parameters with different intrinsic governing factors may lead to over-interpret of the data. We replaced the diurnally-averaged method with the $O_x$ binned method. The scattering plot of the diurnally-averaged data was replaced with hourly-averaged data. Our analysis of hourly averaged data shown that there is a clear influence of photochemical processes in modifying the absorption of BC-containing particles.

**The main modifications of the revised manuscript are listed as following:**

1) **Figure 3 in the text, the correlation between the diurnally-averaged $F_{in}$ and $O_x$ was replaced with the hourly-averaged data.**

[revised manuscript text omitted]

---

## Author Response (AR5)

**Response to the comments**

We thank the reviewers and editor for their thoughtful and thorough reviews, and thank their efforts to make the paper better. Point-by-point responses to the comments are attached below.

**Response to Reviewer 1**

*I think it looks better, but Fig.6C has not been binned properly (the larger $O_x$ end is not included). I would suggest binning with the same data points not same $O_x$ interval.*

Done. We binned the data with same data points (80 points) in each bin. The Fig. 6(a) to (c) were replaced with the new presentation.

[Figure]

**Figure 6:** Relationship between (a) $\omega$, (b) $\omega_{TD}$, and (c) $E_{abs}$ with $O_x$ concentrations. The gray circles are the measurement data with one-hour time resolution. The hourly averaged $\omega$, $\omega_{TD}$, and $E_{abs}$ were then binned in $O_x$ with same data points (80 points) in each bin. The corresponding mean (solid dot), median (center solid line), lower and upper quartile (boxes) and 10th and 90th percentile (whisker) are shown as the box and whisker plots. The corresponding frequency distribution of each parameters are shown in (d) – (g).

**The corresponding text was modified.**

$E_{abs}$ also rose with higher $O_x$ mixing ratios (Fig. 6 (c) and Fig. S10 in the supplement), but with a different pattern compared to $\omega$ and $\omega_{TD}$. From the scattering plot of the

hourly averaged $E_{abs}$ that binned in $O_x$ , a monotonic growth of $E_{abs}$ with increasing $O_x$ can be observed below 45 ppbv $O_x$ (from 2.1 to 2.56~~, with a growth rate of ~ 0.09 ppbv$^{-1}$60~~57 ppbv, $E_{abs}$ values kept constant (~ 2.54).  The small drop of $E_{abs}$ value for $O_x$ bin larger than 57 ppbv was probably caused by the limited data numbers for $O_x$ larger than 75 ppbv and was statistical insignificance. The most frequently occurring value of $E_{abs}$ for the whole measurement was ~ 1.7.

*I don't think it is necessary to always struggle with getting correlation with Ox, as Ox only plays role at daytime, at night the secondary formation is not photochemical. The absorption enhancement is essentially related to the amount of coatings not the Ox. I would recommend to do a correlation with daytime data only and modify the conclusion a bit, then this paper could be accepted.*

DONE. The daytime (from 06:30 to 18:30 local time) and nighttime (from 18:30 to 06:30 local time) data were separated. Only the daytime correlations were fitted. By comparing the daytime and nighttime results, we can see that photochemistry plays a positive effect on the increment of absorption enhancement.

**The corresponding text was modified.**

Since the emission sources, weather conditions and aging degrees of BC particles varied from day to day, the relationship between $E_{abs}$ and atmospheric chemistry is rather complex. Four selected cases with different wind directions were used to demonstrate this complexity (as shown in Fig. 7, the day- and night-time data were separated). The corresponding wind directions and speeds, RH values, and CO concentrations are shown in Fig. S10 in the supplement. The patterns of $E_{abs}$ with $O_x$ were different for air masses from different directions. For case 1 and 2, the mean values of $E_{abs}$ were comparable (1.9 ± 0.2 for case 1, and 1.8 ± 0.6 for case 2), and the hourly-averaged $\omega$ and $\omega_{TD}$ grew with increasing $O_x$ in both cases.  In case 1, $E_{abs}$ ranged from 1.5 to 2.3, with a growth rate of ~ ~ 0.01 ppbv$^{-1}$ in the  daytime. During this period, winds were typically from the north, which corresponding to a short transported pathway of air masses (Fig. S9 in the supplementary). The low degree of aging led to small $E_{abs}$ value. In case 2, $E_{abs}$ ranged from 1.1 to 3.7, and the corresponding growth rates  was ~ ~ 0.065 ppbv$^{-1}$ in the daytime. It is worth noting that the low $\underline{E_{abs}}$ values in this period corresponded to low $\omega$ values (Fig. 7(c)), which indicated the influence on the local emissions on $\underline{E_{abs}}$. For case 3 and 4, $\omega$ and $\omega_{TD}$ increased slowly with $O_x$ in comparison with case 1 and 2. Monotonic relationships were found here, with growth rates of ~ 0.01 and 0.05 ppbv$^{-1}$, respectively. The large $E_{abs}$ values in the daytime than nighttime suggested that photochemistry played a positive effect on the increment of absorption enhancement. These results demonstrated the complex influence of

emission and aging degree of BC particles in modifying the light absorption of BC-containing particles. For air masses from different directions, the relationship between $E_{abs}$ and $O_x$ may be different.

[Figure]

**Figure 7:** Four selected case study of the variations of hourly-averaged $\omega$, $\omega_{TD}$, and $E_{abs}$ as a function of hourly-averaged $O_x$ concentrations. (a,b) Case 1: on June 16, 2016, winds were typically from the north; (c,d) Case 2: June 25 to 26, 2016, winds direction varied from north to south.; (e,f) Case 3: July 7 to 8, winds were mainly from the southeast; 2016; and (g,h) Case 4: July 9 to 12, 2016, winds were mainly from the northeast. The daytime (from 06:30 to 18:30 local time) and nighttime (from 18:30 to 06:30 local time) data were marked in different colors and symbols. The slope of the linear regression (red and blue lines, only for daytime data) is representative of the oxidation rate of each parameter (the fit standard error is shown in brackets).

**Response to Reviewer #2 comments**

*I am satisfied now how the authors modified the text regarding the association between absorption enhancement and Ox concentration. However, I think the authors still should tone down regarding their claim about the absorption enhancement dependency 
[revised manuscript text omitted]